# Towards Robust Pseudo-Label Learning in Semantic Segmentation: An Encoding Perspective

**Wangkai Li[1], Rui Sun[1], Zhaoyang Li[1], Tianzhu Zhang[1,2]**[*]
[1] University of Science and Technology of China
[2]National Key Laboratory of Deep Space Exploration, Deep Space Exploration Laboratory
`{lwklwk, issunrui, lizhaoyang}@mail.ustc.edu.cn, tzzhang@ustc.edu.cn`

## Abstract

Pseudo-label learning is widely used in semantic segmentation, particularly in label-scarce scenarios such as unsupervised domain adaptation (UDA) and semi-supervised learning (SSL). Despite its success, this paradigm can generate erroneous pseudo-labels, which are further amplified during training due to utilization of one-hot encoding. To address this issue, we propose ECOCSeg, a novel perspective for segmentation models that utilizes error-correcting output codes (ECOC) to create a fine-grained encoding for each class. ECOCSeg offers several advantages. First, an ECOC-based classifier is introduced, enabling model to disentangle classes into attributes and handle partial inaccurate bits, improving stability and generalization in pseudo-label learning. Second, a bit-level label denoising mechanism is developed to generate higher-quality pseudo-labels, providing adequate and robust supervision for unlabeled images. ECOCSeg can be easily integrated with existing methods and consistently demonstrates significant improvements on multiple UDA and SSL benchmarks across different segmentation architectures. Code is available at https://github.com/Woof6/ECOCSeg.

## 1 Introduction

Semantic segmentation has seen significant improvements with recent advances in deep neural networks[55, 9, 17, 16]. However, a major challenge in semantic segmentation is the requirement of a large volume of fine-grained pixel-level labels, which can be time-consuming and labor-intensive to obtain [19]. Due to the readily available nature of image data, unsupervised domain adaptation (UDA) and semi-supervised learning (SSL) have been introduced in semantic segmentation to handle the label-scarce scenarios. UDA involves learning from synthetic labeled data and transferring knowledge to real unlabeled target domains, while SSL utilizes a tiny portion of annotated data to generalize on unseen data. As a result, UDA and SSL are gaining significant attention as promising approaches to reduce the reliance on extensive annotations in semantic segmentation.

In both UDA and SSL settings, models are trained using annotated and unlabeled data simultaneously. Existing mainstream methods introduce common paradigms, which can be grouped into the self-training pipeline and the consistency regularization framework. Specifically, self-training methods [79, 41] leverage a temporally smoothed exponential moving average (EMA) model as a teacher to generate stable pseudo labels for unlabeled data. On the other hand, the consistency regularization methods [72, 3] encourage the model to produce consistent predictions for the same sample across different perturbation views. These paradigms can be summarized as pseudo-label learning, where the network's predictions are used as supervision for unlabeled data.

---

[*]Corresponding author

39th Conference on Neural Information Processing Systems (NeurIPS 2025).

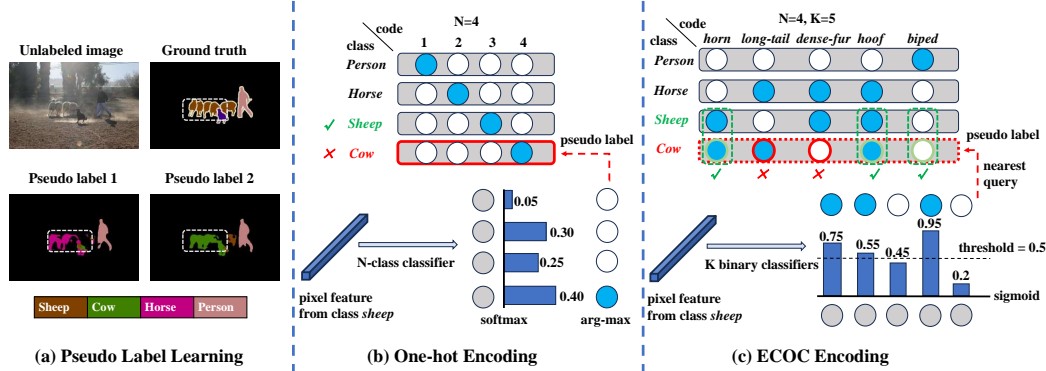

Figure 1: Comparison of two label encoding methods. (a) Examples of erroneous pseudo labels. (b) Existing methods perform pixel-level classification using argmax-based one-hot encoding. (c) The proposed ECOCSeg predicts the multi-bit binary encoding, which disentangles the classes into fine-grained attributes and enhances the stability of the training process in pseudo-label learning.

Although achieving promising results, the inevitable errors in pseudo labels misled the training process. Typical approaches design filter-out mechanisms [93, 72] and only use high-confidence pseudo-labels for training. However, this paradigm tends to make the model focus on learning from easy samples while neglecting difficult ones, resulting in a sub-optimal performance. Another alternative is to utilize weighting functions [36, 77] that assign weights based on confidence of pseudo-labels. While potentially effective, this approach requires careful design and selection of appropriate hyperparameters, which inevitably compromise its applicability. Based on above discussions, we investigate existing works primarily concentrate on developing specific selection strategies for pseudo-labels but rarely consider the impact of the encoding form assigned for classes. As shown in Fig.1 (a), the pixel features of the class *sheep* are being confused by the classifier as *horse* or *cow*, and an erroneous pseudo-label is typically encoded in a one-hot manner through the argmax operation (see Fig.1 (b)). We speculate that similar classes share common visual attributes, leading to confusing pseudo-labels and further misguiding the training process. *How to utilize the shared attributes among confusing classes to design a suitable encoding form for pseudo-label learning is rarely explored.*

To explore the encoding form tailored for pseudo-label learning, we explicitly disentangle the classes into fine-grained attributes and consider each class a set of attributes. As shown in Fig.1 (c), even with incorrect predictions in specific attributes, confusing classes still exhibit shared attribute characteristics. For instance, both *sheep* and *cow* have *horn* and *hoof* and are not *biped*. Despite potential misclassification, accurate prediction of these shared attributes can still provide valuable guidance for effectively training the network. Based on this observation, we resort to error-correcting output codes (ECOC) [21] to assign a binary bit string (codeword) as an encoding for each class, decomposing the $N$-class classification problem into $K$ two-class subtasks. The collection of codewords corresponding to each class forms a codebook. This paradigm determines the class by predicting a $K$-bit binary encoding and selecting the nearest neighbor query in the codebook. The encoding form created by suitable ECOC enjoys two properties: **class discriminability**, ensuring well-separated classes by sufficient Hamming distance between codewords, and **attribute diversity**, ensured by making each bit-position classifier uncorrelated. ECOC encoding endows the model with the ability to handle partial inaccurate bits and make classification decisions. With a theoretical guarantee (Sec. 4), we show that ECOC can serve as an effective equivalent to one-hot encoding in fully supervised settings, and exhibits greater robustness in pseudo-label learning by achieving a tighter classification error bound under a sufficiently large minimum code distance.

In this paper, we propose ECOCSeg, a novel segmentation framework designed for pseudo-label learning. ECOCSeg leverages error-correcting output codes (ECOC) as the class representation and creates fine-grained encoding forms to denoise pseudo-labels. Compared to the widely adopted pseudo-label learning paradigm, which involves encoding form (typically one-hot), pseudo-label selection strategy (typically weighting), and optimization criteria (typically cross-entropy loss), ECOCSeg introduces innovations tailored to the challenges of pseudo-label learning. (1) ***ECOC-based Encoding Form***. To implement ECOC as an alternative to the typically argmax-based one-hot

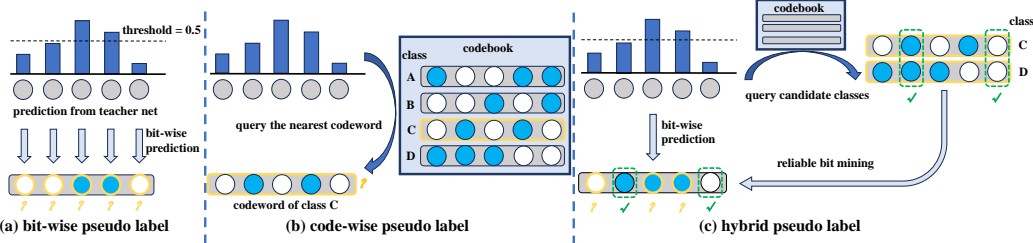

Figure 2: Demonstration of different forms of assigning pseudo labels introduced by ECOCSeg.

encoding in the segmentation paradigm, we explore two simple yet effective coding strategies, i.e., **max-min distance encoding** and **text-based encoding**, to consider robustness and the relationship between classes, respectively. (2) *Bit-level Denoising Mechanism*. To consider the noise in pseudo labels, we present two assigning forms: **bit-wise** pseudo label and **code-wise** pseudo label (Fig. 2 (a) and (b)). The former provides softer supervision by quantifying the output into bit-level codes, while the latter queries the nearest codeword from the codebook as pseudo labels, effectively rectifying inaccurate bits when the classification is accurate but potentially introducing additional noise when incorrect. To leverage the strengths of both forms, we propose a reliable bit mining algorithm to identify candidate classes and determine the shared bits among corresponding codewords as reliable bits, capturing the confidence part of **code-wise** labels. By combining them with **bit-wise** labels, we obtain more robust pseudo-labels in a hybrid way, improving pseudo-label learning stability. (3) *Customized Optimization Criteria*. Intuitively, we can directly use binary cross-entropy for training. However, it optimizes each binary classifier independently, lacking structured representation space constraints and leading to slower convergence. To address this issue, we introduce customized optimization criteria, namely **pixel-code distance** and **pixel-code contrast**. These criteria optimize our framework effectively, equipped with intra-class compactness and inter-class separation, further enhancing overall performance.

Our contributions can be summarized as follows: (1) We present a new perspective to consider pseudo-label noise and propose designing a suitable encoding form for pseudo-label learning that utilizes shared attributes among confusing classes. (2) We formalize pseudo-label learning into three fundamental components for analysis: encoding form, pseudo-label selection strategy, and optimization criteria, and correspondingly develop an ECOC-based encoding form, a bit-level denoising mechanism, and customized loss functions to enhance performance. (3) We theoretically analyze the performance of ECOC and one-hot encoding in both fully supervised and pseudo-label learning settings, demonstrating that with suitable codebook design, ECOC has greater potential to tolerate label noise. (4) We implement ECOCSeg, which can be easily built upon existing pseudo-label learning frameworks and consistently improves performance on multiple UDA and SSL benchmarks.

## 2 Related Work

### 2.1 Label-scarce Semantic Segmentation

Although deep models have achieved remarkable success in various tasks [42, 29, 80, 23, 13, 48, 88, 86], they heavily rely on large amounts of labeled training data and struggle to generalize to data with shifted distributions. This is particularly evident in semantic segmentation, where alternative approaches have been introduced to avoid laborious pixel-wise annotation. Unsupervised domain adaptation (UDA) [27, 26, 81, 78, 15, 64, 32, 31, 14] aims to transfer knowledge from labeled source domains to unlabeled target domains, enabling models to perform well in the target domain despite distribution differences. Semi-supervised learning (SSL) [12, 92, 87, 99, 76, 75] leverages a combination of a few manually annotated target samples with a large pool of unlabeled samples to enhance model performance. Weakly supervised learning [89, 37, 1, 6] addresses the challenge by utilizing less precise annotation signals, such as image-level labels or bounding boxes. Few-shot learning [22, 84, 53, 90, 50, 56, 51, 49] approaches tackle scenarios with a small number of annotated samples by leveraging prior knowledge and meta-learning techniques. UDA and SSL, in particular, share a similar objective of using unlabeled (target) data to improve the performance of models

trained with labeled (source) data only. In this work, we explore these two settings from a unified perspective of pseudo-label learning.

## 2.2 Pseudo-label Learning

Two popular paradigms are often employed when training a model with unlabeled data: self-training-based methods [105, 79, 34, 41, 46] and consistency regularization-based methods [72, 12, 3, 93]. In self-training, the model is trained on unlabeled samples using pseudo labels derived from a teacher network. Consistency regularization aims to ensure prediction stability across different perturbations. Both can be viewed as pseudo-label learning [103, 44, 66]. More recently, state-of-the-art segmentation methods in UDA and SSL combine both technologies [85, 35, 54, 47, 74]. Although making significant progress, incorrect pseudo-labels can mislead the model's training process. Typical approaches incorporate filtering mechanisms [72, 93] to train the model exclusively with highly confident pseudo-labels, while recent research focuses on identifying suitable weight functions to improve training stability [36, 77, 34]. Some works design new optimization criteria inspired by negative learning [66, 87] to improve learning from pseudo-labels. In this paper, we focus on the encoding form of the pseudo-label, which is an orthogonal direction to the above approaches.

## 3 Method

### 3.1 Preliminaries

For the general formulation of UDA and SSL in semantic segmentation, we are given $n_l$ labeled (source) samples $D_l = (x_i^l, y_i^l)_{i=1}^{n_l}$, where $x_i^l$ represents $i$-th image with $y_i^l$ as the corresponding pixel-wise one-hot label covering $N$ classes, and $n_u$ unlabeled (target) samples $D_l = \{x_i^u\}_{i=1}^{n_u}$ with the same label space. The supervised loss $\mathcal{L}^s$ can be calculated on labeled data:

$$\mathcal{L}^s = \frac{1}{n_l} \sum_{i=1}^{n_l} \frac{1}{HW} \sum_{j=1}^{H \times W} \ell_{ce}(F(x_{ij}^l), y_{ij}^l), \tag{1}$$

where $\ell_{ce}$ denotes the cross-entropy loss. The segmentation model, $F$, can be defined as $F = h \circ g$, where $g : \mathcal{X} \to \mathcal{Z}$ lifts each pixel of the input image in $\mathcal{X}$ to the feature space $\mathcal{Z}$ and $h : \mathcal{Z} \to \mathbb{R}^N$ is a pixel-wise classifier to give a score for each class. The unsupervised loss $\mathcal{L}^u$ can be formulated in a unified form of pseudo-label learning as:

$$\mathcal{L}^u = \frac{1}{n_u} \sum_{i=1}^{n_u} \frac{1}{HW} \sum_{j=1}^{H \times W} q(p_{ij}) \ell_{ce}(F(\mathcal{A}^s(x_{ij}^u)), \hat{y}_{ij}^u), \tag{2}$$

$$\hat{y}_{ij}^u = argmax(\hat{F}(\mathcal{A}^w(x_{ij}^u))), \tag{3}$$

where $\hat{y}_{ij}^u$ is pseudo label produced by teacher model $\hat{F}$ and $\mathcal{A}^w/\mathcal{A}^s$ denotes weakly/strongly-augmented strategies. We define $q(p_{ij})$ as a quality estimate conditioned on confidence $p_{ij}$ for pseudo labels, which can be implemented with threshold filtering or a weighting function. The overall objective function is $\mathcal{L} = \mathcal{L}^s + \lambda \mathcal{L}^u$.

### 3.2 Method Overview

An overview of our method ECOCSeg, a pseudo-label learning framework for semantic segmentation, is shown in Fig. 3. We propose an ECOC-based dense classification paradigm (Sec. 3.3), where error-correcting codes create a fine-grained output representation for each class. Then, we study the different forms of pseudo labels driven by ECOCSeg and propose a reliable bit mining algorithm to refine pseudo labels in a bit-level way (Sec. 3.4). Finally, the optimization criteria are developed to train this framework for further enhanced performance (Sec. 3.5).

### 3.3 ECOC-based Dense Classification

Traditionally, semantic segmentation is formulated as a discriminative learning problem based on softmax projection. For each pixel example $i$, the embedding $z_i \in \mathbb{R}^D$ is extracted from $g$ and fed

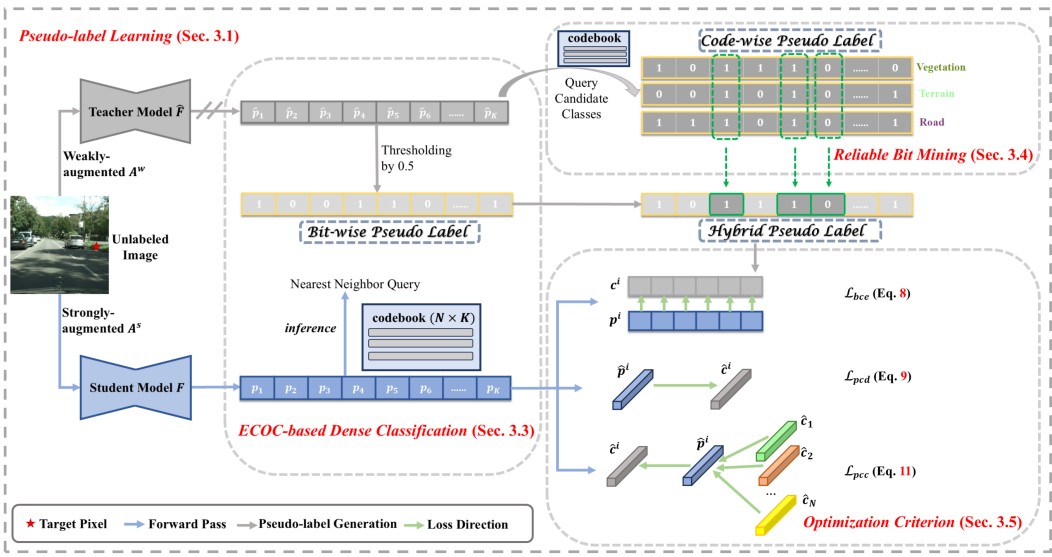

Figure 3: **Pipeline illustration of ECOCSeg**. We introduce a new perspective for semantic segmentation (Sec. 3.3), propose a reliable bit mining algorithm to refine the pseudo label (Sec. 3.4), and develop customized optimization criteria (Sec. 3.5).

into $h$ for $N$-way classification:

$$p(n|\boldsymbol{z}_i) = \frac{exp(\boldsymbol{w}_n^{\mathrm{T}} \boldsymbol{z}_i)}{\sum_{n'=1}^{N} exp(\boldsymbol{w}_n^{\mathrm{T}} \boldsymbol{z}_i)}. \tag{4}$$

In this paradigm, $p(n|\boldsymbol{z}_i) \in [0, 1]$ is the probability that pixel $i$ being assigned to class $n$ and $h$ is a $N$-class classifier parameterized by $\boldsymbol{W} = [\boldsymbol{w}_1^{\mathrm{T}}; \ldots; \boldsymbol{w}_N^{\mathrm{T}}] \in \mathbb{R}^{N \times D}$, in which bias term is omitted.

Our ECOCSeg reformulates the task from a view of dense classification based on error-correcting output codes. An error-correcting codebook is defined as a matrix of binary values with the size of $N \times K$, where each class is represented by a specific codeword of length $K$. The $N$-class classifier is replaced by $K$ binary classifiers, with $p(k|\boldsymbol{z}_i) = sigmoid(\boldsymbol{w}_k^{\mathrm{T}} \boldsymbol{z}_i)$ to predict probability that the $k$-th bit of pixel $i$ being assigned to digit 1. The class is determined by nearest neighbor algorithm within the codebook where soft Hamming distance $d_{SH}$ is used as a metric:

$$d_{SH}(\boldsymbol{c}_n, \boldsymbol{p}^i) = \frac{1}{K} \sum_{k=1}^{K} \|p(k|\boldsymbol{z}_i) - \boldsymbol{c}_{nk}\|_1, \tag{5}$$

$$\hat{n}^i = \arg\min_{n} \{d_{SH}(\boldsymbol{c}_n, \boldsymbol{p}^i)\}, \tag{6}$$

where $\boldsymbol{c}_n$ is the codeword of class $n$, consisting of $\boldsymbol{c}_{nk} \in \{0, 1\}$, $\boldsymbol{p}^i$ is predicted probability vector, consisting of $p(k|\boldsymbol{z}_i) \in [0, 1]$, and $\hat{n}^i$ is the class assigned for pixel $i$.

To enable the proposed paradigm, we construct the codebook using two algorithms: **max-min distance encoding** and **text-based encoding**, to generate the binary matrix $M \in \{0, 1\}^{N \times K}$ and ensure the validity. Please refer to Appendix B for details on the implementation.

## 3.4 Reliable Bit Mining

Since ECOCSeg represents classes as multi-bit binary codes, it naturally leads to the introduction of two forms of pseudo-labels, as shown in Fig. 2. Both forms present varying levels of label noise in different scenarios. Notably, for the code-wise pseudo-label, the noise only comes from incorrect class decisions by Eq. 6, prompting us to explore more candidate classes to mine the reliable bits. As seen in Fig. 3, the ground truth for the target pixel is $terrain$, while the nearest codeword is $vegetation$, thus making the false classification. If we query the $C$-nearest neighbors, the correct class will fall into the candidate set like $\{vegetation, terrain, road\}$ when $C = 3$. Therefore, the shared part within this set, i.e., $P_s(\boldsymbol{c}_{veg.}, \boldsymbol{c}_{ter.}, \boldsymbol{c}_{roa.})$ can be guaranteed to be accurate, allowing us to consider these bits as reliable.

To provide an adaptive value of $C$ for each pixel, we design an effective strategy to determine the candidate classes and mine the reliable bits at the same time (Alg. 1), where we define confidence $q(k|\boldsymbol{z}_i) = max\{p(k|\boldsymbol{z}_i), 1 - p(k|\boldsymbol{z}_i)\}$ for each bit and set a hyperparameter $T$ as upper bound. Then, we obtain the hybrid pseudo label by fusing the bit-wise label ($\boldsymbol{c}_{bit}^i$) and the reliable bits mined from the code-wise label ($\boldsymbol{c}_{code}^i$) with mask $\mathcal{M}^i$:

$$\boldsymbol{c}_{hyb.}^i = \mathcal{M}^i \odot \boldsymbol{c}_{code}^i + (1 - \mathcal{M}^i) \odot \boldsymbol{c}_{bit}^i. \quad (7)$$

A higher $T$ aligns $\boldsymbol{c}_{hyb.}^i$ more closely with the bit-wise way, while a lower $T$ aligns it closer with the code-wise way. The hybrid pseudo-label takes advantage of both forms and introduces less label noise, which provides a low-noise form of supervision for unlabeled images and improves the stability of pseudo-label learning in semantic segmentation.

**Algorithm 1** Reliable bit mining strategy

1: **Input:** probability vector $\boldsymbol{p}^i \in [0, 1]^K$
2: **Output:** mask of the reliable part $\mathcal{M}^i \in \{0, 1\}^K$
3: **Initialize:** code matrix $M$, confidence threshold $T$, candidate set $S_c = \{\}$, $\mathcal{M}^i = \{1\}^K$
4: compute code distance for each class by Eq. 5;
5: sort the code distance and obtain sorted index $\mathcal{I}$;
6: compute confidence $\boldsymbol{q}^i$;
7: **for** $n = 1$ **to** $N$ **do**
8:     add $\boldsymbol{c}_{\mathcal{I}[n]}$ to $S_c$;
9:     compute the shared part $P_s(S_c)$;
10:     update $\mathcal{M}^i$ with bit positions in $P_s(S_c)$;
11:     compute mean confidence $\boldsymbol{q}_m^i$ in $P_s(S_c)$;
12:     **if** $\boldsymbol{q}_m^i > T$ or $\mathcal{M}^i = \{0\}^K$ **then**
13:         **break**;
14:     **end if**
15: **end for**
16: **return** $\mathcal{M}^i$

## 3.5 Optimization Criterion

In ECOCSeg, the multiclass learning problem is decomposed to $K$ binary classification problems, which can be addressed by a binary cross-entropy (BCE) loss:

$$\mathcal{L}_{bce}^i = -\frac{1}{K} \sum_{k=1}^K [\boldsymbol{c}_{(k)}^i \log p(k|\boldsymbol{z}_i) + (1 - \boldsymbol{c}_{(k)}^i) \log(1 - p(k|\boldsymbol{z}_i))], \quad (8)$$

where $\boldsymbol{c}^i$ is the target codeword assigned for pixel $i$. However, only adopting this training objective is insufficient for two reasons. First, Eq. 8 considers the classification of each bit independently, ignoring the relationships between bits and resulting in a lack of intra-class compactness within features extracted by $g$. Second, this bit-level supervision fails to capture inter-class relationships, neglecting inter-class separation. Although Eq. 8 ensures classifier robustness, it lacks structured representation space constraints. To address these issues, we introduce two extra training objectives: pixel-code distance and pixel-code contrast.

**Pixel-code distance.** To regularize representations and reduce intra-class variation, we introduce a compactness-aware loss that minimizes the cosine distance between logits and codewords:

$$\mathcal{L}_{pcd}^i = 1 - \cos(\hat{\boldsymbol{p}}^i, \hat{\boldsymbol{c}}^i), \quad (9)$$

where $\hat{\boldsymbol{p}}^i$ represents the logits predicted by $K$ binary classifiers, and $\hat{\boldsymbol{c}}^i$ is the standardized version of the codeword $\boldsymbol{c}^i$, with $\hat{\boldsymbol{c}}_k^i \in \{-1, 1\}$.

**Pixel-code contrast.** Eq. 9 encourages intra-class similarity without considering inter-class separation. For two codewords $\boldsymbol{c}_1$ and $\boldsymbol{c}_2$, we define the shared part between them as $P_s(\boldsymbol{c}_1, \boldsymbol{c}_2)$ and the distinctive part as $P_d(\boldsymbol{c}_1, \boldsymbol{c}_2)$. The value of $\hat{\boldsymbol{p}}^i$ changes in $P_s(\boldsymbol{c}_1, \boldsymbol{c}_2)$ and $P_d(\boldsymbol{c}_1, \boldsymbol{c}_2)$ has the same impact on the Eq. 9, while the latter part is more significant since $P_s(\boldsymbol{c}_1, \boldsymbol{c}_2)$ does not distinguish between classes. Thus, a pixel-code contrastive learning strategy is introduced:

$$\mathcal{L}_{pcc}^i = - \log \frac{\exp\left(\langle \hat{\boldsymbol{p}}^i, \hat{\boldsymbol{c}}^i \rangle / \tau\right)}{\exp\left(\langle \hat{\boldsymbol{p}}^i, \hat{\boldsymbol{c}}^i \rangle / \tau\right) + \sum_{\hat{\boldsymbol{c}}^- \in \hat{\boldsymbol{c}}^-} \exp\left(\langle \hat{\boldsymbol{p}}^i, \hat{\boldsymbol{c}}^- \rangle / \tau\right)}, \quad (10)$$

where $\langle, \rangle$ is cosine similarity, $\hat{\boldsymbol{\mathcal{C}}}^- = \{\hat{\boldsymbol{c}}_n\}_{n=1}^N / \hat{\boldsymbol{c}}^i$ and $\tau$ is the temperature to control the concentration level. Note that the target codeword $\hat{\boldsymbol{c}}^i$ is not necessarily included in $\{\hat{\boldsymbol{c}}_n\}_{n=1}^N$ if we adopt the form of a bit-wise pseudo label. Furthermore, this loss term can be rewritten as:

$$\mathcal{L}_{pcc}^i = \log(1 + \sum_{\hat{\boldsymbol{c}}^- \in \hat{\boldsymbol{c}}^-} \exp\left(\langle \hat{\boldsymbol{p}}^i, \hat{\boldsymbol{c}}^- - \hat{\boldsymbol{c}}^i \rangle / \tau\right)), \quad (11)$$

which is only calculated on the $P_d(\hat{\boldsymbol{c}}^i, \hat{\boldsymbol{c}}^-)$ to distinguish between codewords. These two loss terms complement each other to enhance the representative capacity of the learning features. Then, the

segmentation model is trained with combinatorial loss over all training pixel samples:

$$\mathcal{L}_{total} = \mathcal{L}_{bce} + \lambda_1 \mathcal{L}_{pcd} + \lambda_2 \mathcal{L}_{pcc}, \tag{12}$$

which can be utilized in both supervised loss $\mathcal{L}^s$ and unsupervised loss $\mathcal{L}^u$ in pseudo-label learning.

## 4 Theory

In this section, we present our main theoretical results on the performance and robustness of ECOC-based DNNs. We first introduce some key tools and concepts used in our analysis, including the Neural Tangent Kernel (NTK) [38] and its properties in the infinite width limit. Then we state our theorems, which characterize the behavior of ECOC compared to one-hot encoding in the fully supervised setting and the pseudo-label learning setting, respectively. The complete proofs with technical assumptions are deferred to the Appendix A.

**Theorem 4.1** (ECOC Performance in Fully Supervised Setting). *Suppose the ECOC encoding matrix $E([C])$ is nearly orthogonal, i.e., $|E([C])^T E([C]) - nI| \leq \delta$ for some small $\delta > 0$, where $n$ is the code length and $I$ is the identity matrix. Then the ECOC-based DNN achieves performance equivalent to the one-hot encoding in the fully supervised setting, up to an error term depending on $\delta$.*

*Remark* 1. In practice, the codewords of ECOC are often designed to be approximately orthogonal. Due to the smoothness of the NTK, we can expect the performance of ECOC-based DNN to be close to that of one-hot encoding when the codewords are nearly orthogonal.

**Theorem 4.2** (ECOC Robustness in Pseudo-Label Learning). *Suppose $E([C]) \in \{-1, +1\}^{C \times n}$ has code length $n$ and minimum distance $d$, and the binary classifiers $f_1(x), \ldots, f_n(x)$ satisfy the margin condition with parameters $\gamma_1, \ldots, \gamma_n$. Assume that the pseudo-labels are treated as class labels that are corrupted by random noise with probability $\epsilon$. If the minimum distance $d$ satisfies:*

$$d > \frac{16\epsilon\kappa^2}{\gamma^2}\left(\frac{(1+\log 2)n}{2} - \log(2C)\right) + 2\frac{\hat{\gamma}^2}{\gamma^2}, \tag{13}$$

*where $\gamma = \min_k \gamma_k$, $\hat{\gamma} = \min_j \hat{\gamma}_j$, and $\kappa = \kappa(B, L, \phi(0))$, then the classification error probability of the ECOC-based DNNs admits a tighter upper bound than that of one-hot encoding under the same noise level $\epsilon$.*

*Remark* 2. Theorem 4.2 provides a comparison between the robustness of ECOC and one-hot encoding under label noise in the context of pseudo-label learning. It shows that ECOC can achieve a tighter error bound than one-hot encoding, provided that the minimum distance of the ECOC matrix is sufficiently large compared to the noise level. This result suggests that we can employ a larger minimum distance $d$ to cope with higher noise levels, demonstrating the robustness of the ECOC-based DNN against label noise. In our experiments, we also demonstrate that ECOC can achieve better model calibration (Appendix K), thus enhancing the reliability of pseudo-labels.

## 5 Experiment

### 5.1 Experimental Setup

**Datasets.** We evaluate our approach on two standard benchmarks for synthetic-to-real adaptation of street scenes in the UDA task. The synthetic datasets include GTAv [67] (24,966 images) and SYNTHIA [69] (9,400 images). Cityscapes [19], a real-world urban dataset, serves as the target domain, with 2,975 training and 500 validation images. For the SSL setting, we use Cityscapes, PASCAL VOC 2012 [25], a generic object segmentation benchmark with 1,464 training and 1,449 validation images, along with an augmented set of 10,582 additional training images, and COCO [52], a challenging benchmark composed of 118k/5k training/validation images with 81 classes.

**UDA Setting.** We evaluate ECOCSeg on three widely used frameworks, DACS [79] with ResNet101 [29] backbone, DAFormer [34], and MIC [35], with MIT-B5 [91] backbone. Experiments are conducted on one RTX-3090 GPU for DACS and DAFormer, and two for MIC. The network is trained for 40K iterations (batch size 2) using AdamW optimizer with learning rates of $6 \times 10^{-5}$ (encoder) and $6 \times 10^{-4}$ (decoder), weight decay of 0.01, and linear warm-up for the first 1.5K iterations. Images are rescaled and randomly cropped to $512 \times 512$ following DAFormer's augmentation, and the EMA coefficient for updating the teacher net is 0.999.

Table 1: UDA performance on two synthetic-to-real benchmarks, where the IoU improved by ECOCSeg is marked as **bold**. For each benchmark, results are acquired based on CNN-based model [8] (C) and Transformer-based model [91] (T). mIoUs on SYN.→CS. are calculated over 16 classes.

| Method | Arch. | Road | Sidewalk | Building | Wall | Fence | Pole | Light | Sign | Veg | Terrain | Sky | Person | Rider | Car | Truck | Bus | Train | Motor | Bike | mIoU |
|---|---|---|---|---|---|---|---|---|---|---|---|---|---|---|---|---|---|---|---|---|---|
| GTAv→Cityscapes(Val.) | | | | | | | | | | | | | | | | | | | | | |
| ProDA [100] | C | 87.8 | 56.0 | 79.7 | 46.3 | 44.8 | 45.6 | 53.5 | 53.5 | 88.6 | 45.2 | 82.1 | 70.7 | 39.2 | 88.8 | 45.5 | 50.4 | 1.0 | 48.9 | 56.4 | 57.5 |
| CPSL [45] | C | 92.3 | 59.5 | 84.9 | 45.7 | 29.7 | 52.8 | 61.5 | 59.5 | 87.9 | 41.6 | 85.0 | 73.0 | 35.5 | 90.4 | 48.7 | 73.9 | 26.3 | 53.8 | 53.9 | 60.8 |
| TransDA [11] | T | 94.7 | 64.2 | 89.2 | 48.1 | 45.8 | 50.1 | 60.2 | 40.8 | 90.4 | 50.2 | 93.7 | 76.7 | 47.6 | 92.5 | 56.8 | 60.1 | 47.6 | 49.6 | 55.4 | 63.9 |
| ADFormer [30] | T | 96.7 | 75.1 | 88.8 | 57.5 | 45.9 | 45.6 | 55.4 | 59.8 | 90.2 | 45.6 | 92.1 | 70.8 | 43.0 | 91.0 | 78.9 | 79.3 | 68.7 | 52.7 | 65.0 | 69.2 |
| CDAC [83] | T | 97.1 | 78.7 | 91.8 | 59.6 | 57.1 | 59.1 | 66.1 | 72.2 | 91.8 | 53.1 | 94.5 | 79.4 | 51.6 | 94.6 | 84.9 | 87.8 | 78.7 | 64.9 | 67.6 | 75.3 |
| DACS [79] | C | 89.9 | 39.7 | 87.9 | 39.7 | 39.5 | 38.5 | 46.4 | 52.8 | 88.0 | 44.0 | 88.8 | 67.2 | 35.8 | 84.5 | 45.7 | 50.2 | 0.2 | 27.3 | 34.0 | 52.1 |
| +ECOCSeg | C | **95.6** | **71.8** | **90.2** | 37.8 | 31.4 | **44.8** | **50.8** | **58.8** | **90.4** | **50.3** | **91.3** | **68.6** | 23.5 | **91.2** | **49.8** | **55.4** | **8.8** | 15.2 | 9.8 | 54.5↑2.4 |
| DAFormer [34] | T | 95.7 | 70.2 | 89.4 | 53.5 | 48.1 | 49.6 | 55.8 | 59.4 | 89.9 | 47.9 | 92.5 | 72.2 | 44.7 | 92.3 | 74.5 | 78.2 | 65.1 | 55.9 | 61.8 | 68.3 |
| +ECOCSeg | T | **96.7** | **75.6** | 89.4 | **54.0** | **51.4** | **55.1** | 59.4 | **61.9** | **90.1** | 46.6 | 90.0 | 71.5 | 42.4 | **92.8** | **79.7** | **85.4** | **79.1** | **60.0** | 58.2 | 70.5↑2.2 |
| MIC [35] | T | 97.4 | 80.1 | 91.7 | 61.2 | 56.9 | 59.7 | 66.0 | 71.3 | 91.7 | 51.4 | 94.3 | 79.8 | 56.1 | 94.6 | 85.4 | 90.3 | 80.4 | 64.5 | 68.5 | 75.9 |
| +ECOCSeg | T | **97.9** | **81.4** | **91.9** | **62.2** | 54.3 | **64.2** | **67.4** | **76.1** | **92.9** | **54.4** | 94.2 | **82.1** | 53.0 | **95.2** | **89.6** | **90.8** | **82.3** | 61.9 | **69.4** | 76.9↑1.0 |
| SYNTHIA→Cityscapes(Val.) | | | | | | | | | | | | | | | | | | | | | |
| ProDA [100] | C | 87.8 | 45.7 | 84.6 | 37.1 | 0.6 | 44.0 | 54.6 | 37.0 | 88.1 | - | 84.4 | 74.2 | 24.3 | 88.2 | - | 51.1 | - | 40.5 | 45.6 | 55.5 |
| CPSL [45] | C | 87.2 | 43.9 | 85.5 | 33.6 | 0.3 | 47.7 | 57.4 | 37.2 | 87.8 | - | 88.5 | 79.0 | 32.0 | 90.6 | - | 49.4 | - | 50.8 | 59.8 | 57.9 |
| TransDA [11] | T | 90.4 | 54.8 | 86.4 | 31.1 | 1.7 | 53.8 | 61.1 | 37.1 | 90.3 | - | 93.0 | 71.2 | 25.3 | 92.3 | - | 66.0 | - | 44.4 | 49.8 | 59.3 |
| ADFormer [30] | T | 91.8 | 53.6 | 87.0 | 40.5 | 5.2 | 46.8 | 52.1 | 54.9 | 88.4 | - | 92.6 | 72.5 | 45.7 | 86.1 | - | 61.6 | - | 50.4 | 64.4 | 62.1 |
| CDAC [83] | T | 93.1 | 68.5 | 89.8 | 51.2 | 8.9 | 59.4 | 65.5 | 65.3 | 84.7 | - | 94.4 | 81.2 | 57.0 | 90.5 | - | 56.9 | - | 66.8 | 66.4 | 68.7 |
| DACS [79] | C | 80.6 | 25.1 | 81.9 | 21.5 | 2.9 | 37.2 | 22.7 | 24.0 | 83.7 | - | 90.8 | 67.6 | 38.3 | 82.9 | - | 38.9 | - | 28.5 | 47.6 | 48.3 |
| +ECOCSeg | C | **88.0** | 17.6 | **88.2** | 17.3 | **9.3** | **41.7** | **47.4** | **50.2** | **87.8** | - | 89.1 | **72.6** | **41.5** | **86.2** | - | 9.3 | - | **34.5** | **53.6** | 52.1↑3.8 |
| DAformer [34] | T | 84.5 | 40.7 | 88.4 | 41.5 | 6.5 | 50.0 | 55.0 | 54.6 | 86.0 | - | 89.8 | 73.2 | 48.2 | 87.2 | - | 53.2 | - | 53.9 | 61.7 | 60.9 |
| +ECOCSeg | T | **90.6** | **50.3** | **89.1** | **41.8** | **11.3** | 49.5 | **56.8** | **58.3** | **86.9** | - | **91.9** | **76.2** | 44.2 | **88.4** | - | **61.3** | - | **57.8** | 58.3 | 63.3↑2.4 |
| MIC [35] | T | 86.6 | 50.5 | 89.3 | 47.9 | 7.8 | 59.4 | 66.7 | 63.4 | 87.1 | - | 94.6 | 81.0 | 58.9 | 90.1 | - | 61.9 | - | 67.1 | 64.3 | 67.3 |
| +ECOCSeg | T | **94.3** | **68.8** | 89.0 | 42.3 | **13.6** | **60.5** | **68.8** | 57.5 | **90.4** | - | 94.4 | 80.1 | 54.5 | **90.7** | - | **68.7** | - | 64.0 | **67.1** | 69.0↑1.7 |

**SSL Setting.** We implement our method on ST++ [95], FixMatch [72], UniMatch [93] and adopt DeepLabv3+ [10] with a ResNet [29] backbone as our segmentation model. For Pascal, we use a crop size of $321 \times 321$ and $513 \times 513$, a batch size of 8, and a learning rate of 0.001 with an SGD optimizer. The model is trained for 80 epochs using a poly learning rate scheduler on $2\times$ RTX 3090 GPUs. More experiment settings are detailed in Appendix G.

**ECOCSeg Parameters.** ECOCSeg uses $M_{text}$ as the default codebook with codeword length $K = 40$ for Cityscapes and Pascal, and $K = 60$ for COCO. We set the loss weight $\lambda_1 = 5$ and $\lambda_2 = 2$ with the temperature $\tau = 0.5$. The confidence threshold $T$ for reliable bit mining is set to 0.95. Specifically, we use the mean value of bit-wise confidence, i.e., $\frac{1}{K} \sum_{k=1}^{K} q(k|\boldsymbol{z}_i)$, to estimate pixel-wise confidence, which is needed in Eq. 2.

## 5.2 ECOCSeg for UDA

We integrate ECOCSeg with three baselines and compare with state-of-the-art UDA approaches on GTAv→Cityscapes and SYNTHIA→Cityscapes benchmarks. For a fair comparison, we train the model with same hyperparameters as the baseline methods. We report results based on whole inference on DACS, DAFormer and slide inference on MIC without other test time augmentation strategies. As shown in Table 1, ECOCSeg achieves 2.4% and 2.9% gains on the two benchmarks built with DACS and 2.2% and 2.4% gains built with the strong baseline DAFormer. Compared to the previous state-of-the-art method MIC, ECOCSeg also achieves consistent improvements for most classes, resulting in 1.0% and 1.7% gains.

Furthermore, significant gains are primarily observed in confusing classes (e.g., {*road*, *sidewalk*}, {*truck*, *bus*, *train*}), which typically encounter unstable adaptation in pseudo-label learning based on one-hot encoding. This observation is also reflected in qualitative results (Appendix M). While previous methods struggle to distinguish confusing classes, ECOCSeg significantly improves their accuracy, primarily attributed to the supervision provided by higher-quality pseudo-labels.

## 5.3 ECOCSeg for SSL

We evaluate the performance using 1/16, 1/8, and 1/4 labeled data with ResNet-50 and ResNet-101 backbones with three different SSL frameworks. As shown in Table 2, ECOCSeg consistently outperforms the baselines under different partition protocols, training resolutions, and backbone architectures, with gains ranging from 1.1% to 3.7%. This confirms that these pseudo-label learning methods can benefit from the robust pseudo-labels provided by ECOCSeg.

In Appendix G, we evaluate on more powerful baselines, provide more quantitative results on the Cityscapes, COCO and additional real-world Scenarios. ECOCSeg demonstrates significant gains across multiple different datasets and network architectures, indicating the versatility of our method.

Table 2: SSL performance on Pascal. The 321 and 513 denote the training resolution.

| Method | Res. | ResNet-50 | | | ResNet-101 | | |
|---|---|---|---|---|---|---|---|
| | | 1/16 | 1/8 | 1/4 | 1/16 | 1/8 | 1/4 |
| Sup-only | 321 | 61.2 | 67.3 | 70.8 | 65.6 | 70.4 | 72.8 |
| CAC [43] | 321 | 70.1 | 72.4 | 74.0 | 72.4 | 74.6 | 76.3 |
| ST++ [95] | 321 | 72.6 | 74.4 | 75.4 | 74.5 | 76.3 | 76.6 |
| +ECOCSeg | 321 | **74.0** | **76.1** | **76.5** | **77.1** | **77.9** | **78.0** |
| UniMatch [93] | 321 | 74.5 | 75.8 | 76.1 | 76.5 | 77.0 | 77.2 |
| +ECOCSeg | 321 | **76.4** | **77.5** | **77.6** | **78.1** | **78.6** | **78.9** |
| Sup-only | 513 | 62.4 | 68.2 | 72.3 | 67.5 | 71.1 | 74.2 |
| $U^2$PL [87] | 513 | 72.0 | 75.1 | 76.2 | 74.4 | 77.6 | 78.7 |
| PS-MT [54] | 513 | 72.8 | 75.7 | 76.4 | 75.5 | 78.2 | 78.7 |
| DAW [77] | 513 | 76.2 | 77.6 | 77.4 | 78.5 | 78.9 | 79.6 |
| RankMatch [58] | 513 | 76.6 | 77.8 | 78.3 | 78.9 | 79.2 | 80.0 |
| Fixmatch [72] | 513 | 70.6 | 73.9 | 75.1 | 74.3 | 76.3 | 76.9 |
| +ECOCSeg | 513 | **74.3** | **75.5** | **76.3** | **76.0** | **77.8** | **78.2** |
| UniMatch [93] | 513 | 75.8 | 76.9 | 76.8 | 78.1 | 78.4 | 79.2 |
| +ECOCSeg | 513 | **77.1** | **78.3** | **78.5** | **79.2** | **79.8** | **80.3** |

Table 3: Ablation study on optimization criterion, built with DAFormer under the **fully supervised learning setting** on Cityscapes.

| Encoding | $\mathcal{L}_{ce}$ | $\mathcal{L}_{bce}$ | $\mathcal{L}_{pcd}$ | $\mathcal{L}_{pcc}$ | mIoU |
|---|---|---|---|---|---|
| one-hot | ✓ | - | - | - | 77.6 |
| | - | ✓ | - | - | 76.3 |
| | - | ✓ | ✓ | - | 77.9 |
| $M_{text}$ | - | ✓ | - | ✓ | 77.8 |
| | - | ✓ | ✓ | ✓ | **78.1** |
| $M_{mmd}$ | - | ✓ | ✓ | ✓ | 77.7 |

Table 4: Ablation study on confidence threshold $T$, built with DAFormer under the **UDA setting** on GTAv→Cityscapes.

| Encoding | baseline | code | bit | 0.9 | 0.95 | 0.99 | oracle |
|---|---|---|---|---|---|---|---|
| $M_{mmd}$ | 68.3 | 69.0 | 69.6 | 69.4 | 69.9 | **70.2** | 77.7 |
| $M_{text}$ | 68.3 | 69.7 | 69.4 | 70.0 | **70.5** | 69.8 | 78.1 |

## 5.4 Diagnostic Experiment

**Optimization Criterion.** We evaluate the proposed optimization criterion under the fully supervised setting using DAFormer on Cityscapes in Table 3. The first line denotes the original argmax-based one-hot encoding supervised by cross-entropy loss. When implemented only with $\mathcal{L}_{bce}$, the accuracy for $M_{text}$ is lower than baseline due to sub-optimal learning by independent binary classification. Adding $\mathcal{L}_{pcd}$ or $\mathcal{L}_{pcc}$ individually brings gains with 1.6% and 1.5%. Combining all the losses yields the best performance, surpassing the one-hot paradigm by a higher margin of 0.5%. When implemented with $M_{mmd}$, the performance slightly degrades but still achieves competitive results.

**Confidence Threshold $T$.** We conduct a study comparing pseudo-label performance using code-wise and bit-wise forms in Table 4. The former performs better when combined with $M_{text}$, while the latter achieves better results when combined with $M_{mmd}$. This is because the code-wise form introduces more noise with larger code distance (please refer to Appendix C for more analysis). The hybrid form combines the advantages of both forms to achieve consistent improvements. It is worth noting that despite the lower oracle performance (fully supervised setting) with $M_{mmd}$, it achieves competitive domain adaptation capability, benefiting from robust pseudo-label learning. Furthermore, the confidence threshold $T$ controls the mixing ratio in hybrid labels as discussed in Sec. 3.4: when $T = 0.5$, it is equivalent to code-wise form, while $T = 1$ is equivalent to bit-wise form. To further investigate the impact of $T$, we quantify the count of differing bits between the code-wise labels and bit-wise labels (Difference Count), and the count of bits corrected by the reliable bit mining algorithm (Correction Count) in Fig. 4 (a). Across different values of $T$, as the training process progresses, the differences between the two pseudo-label forms decrease, and the count of corrected bits increases. An appropriate selection of $T$ value can generate a robust hybrid label that combines the advantages of both forms, thereby reducing pseudo-label noise.

**Analysis of Reliable Bit Mining.** Fig. 4 (b) shows an example of the reliable bit mining algorithm during training. The model first queries the nearest class and calculates the confidence map. Due to the confusion between *sidewalk* and *road* classes in the marked area, the confidence of this area falls below the threshold $T$. When querying the second class, the model obtains accurate classification, and the shared bits between code-wise labels of the two classes exhibit higher confidence, which can be viewed as reliable bits.

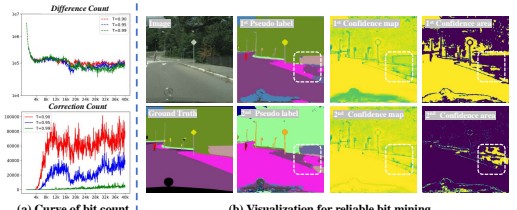

Figure 4: (a) Bit count curves under different $T$. (b) Visualization of 2-nearest codewords and confidence map; dotted boxes indicate confusing areas.

As classification errors tend to occur in low confidence regions, the bit-wise pseudo-label provides softer supervision with lower noise compared to the code-wise form. This hybrid approach combines the advantages of both forms, allowing for a more robust and effective training process.

# 6   Conclusion

In this paper, we present ECOCSeg, a novel framework that introduces a new perspective that utilizes the error-correcting output codes as a fine-grained encoding form for each class and facilitates stable pseudo-label learning in semantic segmentation. By leveraging an ECOC-based encoding form, a bit-level pseudo-label denoising mechanism, and customized optimization criteria, ECOCSeg effectively addresses the challenges associated with pseudo-label learning. The versatility of ECOCSeg allows it to be easily integrated with existing works, consistently demonstrating improvements across multiple unsupervised domain adaptation (UDA) and semi-supervised learning (SSL) benchmarks.

## Acknowledgements

This work was partially supported by the National Key R&D Program of China (Grant No. 2024YFB3909902), and the Youth Innovation Promotion Association of the Chinese Academy of Sciences (CAS).

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

# A  Proofs of the Theoretical Results

In this appendix, we provide the complete proofs of the theorems presented in the main text. We first state the assumptions and lemmas used in our analysis, and then present the detailed proofs.

**Assumption A.1** (Lipschitz continuity). The activation function $\phi$ is B-Lipschitz continuous, i.e., $|\phi(x) - \phi(y)| \leq B|x - y|$ for all $x, y \in \mathbb{R}$.

**Assumption A.2** (Bounded inputs). The input data $x$ satisfies $||x||_2 \leq 1$.

**Assumption A.3** (NTK assumptions). The Neural Tangent Kernel (NTK) converges to a deterministic kernel in the infinite width limit, and the NTK matrix $K(X, X)$ is positive definite, where $X$ is the training data matrix.

**Assumption A.4** (Initialization). The weights and biases of the DNN are initialized according to a standard Gaussian distribution with appropriate scaling.

**Lemma A.5** (NTK convergence, [38]). *Under Assumptions A.1-A.4, the NTK of a DNN converges in probability to a deterministic kernel $K$ as the width of the hidden layers goes to infinity.*

**Lemma A.6** (Hidden Layer Output Bound, [96]). *Let Assumptions A.1-A.4 hold. Then, for any hidden layer $l \leq L - 1$ and any $\delta > 0$,*

$$\frac{|x^{(l)}|_2}{\sqrt{n_l}} \leq \left(1 - \frac{B + |\phi(0)|}{1 - B}\right) B^l + \frac{B + |\phi(0)|}{1 - B} + \delta \tag{14}$$

*holds with probability at least $1 - \delta$ when $n$ is large enough, where the hidden layer width $n_l = \alpha_l n$ with constant $\alpha_l > 0$ for all $1 \leq l \leq L - 1$.*

Now we prove the theorems.

***Proof of Theorem 4.1***. Let $f_{\text{ECOC}}$ and $f_{\text{one-hot}}$ denote the functions represented by the ECOC-based DNN and the one-hot encoding DNN, respectively. By Lemma A.5, in the infinite width limit, the outputs of these DNNs can be expressed using the Neural Tangent Kernel (NTK) as:

$$f_{\text{ECOC}}(x) = K(X, x)(K(X, X) + \lambda I)^{-1} E([C])\tilde{Y}, \tag{15}$$

$$f_{\text{one-hot}}(x) = K(X, x)(K(X, X) + \lambda I)^{-1}\tilde{Y}, \tag{16}$$

where $\tilde{Y}$ denotes the one-hot encoded target, and the ECOC target is $Y = E([C])\tilde{Y}$. Then, the decoding process for the ECOC-based DNN is:

$$\begin{aligned} D(f_{\text{ECOC}}(x)) &= \arg\min_{c \in [C]} ||E([C])e_c - K(X, x)(K(X, X) + \lambda I)^{-1} E([C])\tilde{Y}||_2 \\ &= \arg\min_{c \in [C]} ||e_c - \tilde{Y}(K(X, X) + \lambda I)^{-1} K(X, x)||_{E([C])^T E([C])}. \end{aligned} \tag{17}$$

where $e_c \in \mathbb{R}^C$ is the $c$-th one-hot codeword and $|x|_A \triangleq \sqrt{x^T A x}$ is the Mahalanobis norm with positive definite matrix $A$. For the one-hot encoding DNN, the decoding process is:

$$D(f_{\text{one-hot}}(x)) = \arg\min_{c \in [C]} \left\| e_c - \tilde{Y}(K(X, X) + \lambda I)^{-1} K(X, x) \right\|_2. \tag{18}$$

When $E([C])$ is orthogonal, i.e., $E([C])^T E([C]) = nI$, the Mahalanobis norm reduces to the scaled Euclidean norm:

$$\|x\|_{E([C])^T E([C])} = \sqrt{n}\|x\|_2. \tag{19}$$

In this case, the decoding processes for ECOC and one-hot encoding are equivalent up to a scaling factor, leading to the same classification results. When $E([C])$ is nearly orthogonal, i.e., $|E([C])^T E([C]) - nI| \leq \delta$ for some small $\delta > 0$, the Mahalanobis norm is a perturbed version of the scaled Euclidean norm. The difference in the decoding metrics leads to a difference in the classification performance. Specifically, let $\hat{c}_{\text{ECOC}}$ and $\hat{c}_{\text{one-hot}}$ be the predicted class labels from

the ECOC-based and one-hot encoding DNNs, respectively. Then,

$$
\begin{aligned}
\mathbb{P}(\hat{c}_{\text{ECOC}} \neq \hat{c}_{\text{one-hot}}) = \mathbb{P}(\exists c \neq \hat{c}_{\text{one-hot}} : &|e_c - \tilde{Y}(K(X,X)+\lambda I)^{-1}K(X,x)|E([C])^T E([C]) \\
&\leq |e_{\hat{c}_{\text{one-hot}}} - \tilde{Y}(K(X,X)+\lambda I)^{-1}K(X,x)|E([C])^T E([C])) \\
\leq \sum_{c \neq \hat{c}_{\text{one-hot}}} \mathbb{P}(&|e_c - \tilde{Y}(K(X,X)+\lambda I)^{-1}K(X,x)|E([C])^T E([C]) \\
&\leq |e_{\hat{c}_{\text{one-hot}}} - \tilde{Y}(K(X,X)+\lambda I)^{-1}K(X,x)|E([C])^T E([C])) \\
\leq \sum_{c \neq \hat{c}_{\text{one-hot}}} \mathbb{P}(&\sqrt{n}||e_c - \tilde{Y}(K(X,X)+\lambda I)^{-1}K(X,x)||_2 - \delta \\
&\leq \sqrt{n}||e_{\hat{c}_{\text{one-hot}}} - \tilde{Y}(K(X,X)+\lambda I)^{-1}K(X,x)||_2 + \delta) \\
= \sum_{c \neq \hat{c}_{\text{one-hot}}} \mathbb{P}((&||e_c - \tilde{Y}(K(X,X)+\lambda I)^{-1}K(X,x)||_2 - ||e\hat{c}_{\text{one-hot}} - \\
&\tilde{Y}(K(X,X)+\lambda I)^{-1}K(X,x)||_2) \leq \frac{2}{\sqrt{n}}\delta)
\end{aligned}
\tag{20}
$$

let $\Delta_c = ||e_c - \tilde{Y}(K(X,X)+\lambda I)^{-1}K(X,x)||_2$. By definition of $\hat{c}_{\text{one-hot}}$, $\Delta\hat{c}_{\text{one-hot}} \leq \Delta_c$ for all $c \neq \hat{c}_{\text{one-hot}}$. Therefore,

$$
\mathbb{P}(\hat{c}_{\text{ECOC}} \neq \hat{c}_{\text{one-hot}}) \leq \sum_{c \neq \hat{c}_{\text{one-hot}}} \mathbb{P}(\Delta_c - \Delta\hat{c}_{\text{one-hot}} \leq \frac{2}{\sqrt{n}}\delta)
\tag{21}
$$

When $E([C])$ is orthogonal, i.e., $\delta = 0$, we have $\mathbb{P}(\hat{c}_{\text{ECOC}} \neq \hat{c}_{\text{one-hot}}) = 0$. When $E([C])$ is $\delta$-nearly orthogonal, $|\Delta_c - \Delta_{\hat{c}_{\text{one-hot}}}| \leq \frac{2}{\sqrt{n}}\delta$ for all $c$. Thus,

$$
\mathbb{P}(\hat{c}_{\text{ECOC}} \neq \hat{c}_{\text{one-hot}}) \leq (C-1) \cdot \mathbf{1}_{\frac{2}{\sqrt{n}}\delta \geq \min_{c \neq \hat{c}_{\text{one-hot}}} \Delta_c - \Delta_{\hat{c}_{\text{one-hot}}}}.
$$

If $\delta = o(\frac{1}{\sqrt{n}})$, then $\frac{2}{\sqrt{n}}\delta \to 0$ as $n \to \infty$, while $\min_{c \neq \hat{c}_{\text{one-hot}}} \Delta_c - \Delta_{\hat{c}_{\text{one-hot}}}$ converges to a positive constant. Therefore, the indicator function becomes 0 for sufficiently large $n$, making $\mathbb{P}(\hat{c}_{\text{ECOC}} \neq \hat{c}_{\text{one-hot}})$ arbitrarily small. The ECOC-based DNN achieves performance equivalent to the one-hot encoding DNN up to an error term depending on $\delta$. $\qquad\square$

**Lemma A.7** (Concentration of noisy functions). *Let $f$ be a real-valued function on a probability space $(\mathcal{X}, \mathcal{A}, P)$ such that $\mathbb{E}[f^2] < \infty$. Let $\tilde{f}$ be a noisy version of $f$ such that $\mathbb{E}[(\tilde{f}(x) - f(x))^2] \leq \sigma^2$ for all $x \in \mathcal{X}$. Then for any $\delta > 0$,*

$$
P\left(\sup_{x \in \mathcal{X}} |\tilde{f}(x) - \mathbb{E}[\tilde{f}(x)]| > \sigma\sqrt{2\log(2/\delta)}\right) \leq \delta.
\tag{22}
$$

**Lemma A.8** (Binary classifier perturbation bound). *Under the margin condition and the random label noise model, for any $\delta > 0$, with probability at least $1 - \delta$, the error probability of each binary classifier $f_k(x)$ is bounded by*

$$
p_k \leq 2\exp\left(-\frac{\gamma_k^2}{8\epsilon_k(\kappa(B, L, \phi(0)) + \delta/2)^2}\right),
\tag{23}
$$

*where $\epsilon_k$ is the label noise probability for the $k$-th bit, $B$ is the Lipschitz constant of the activation function, $L$ is the depth of the network, and $\gamma_k$ is margin.*

*Proof.* We first review the conditions and notations in the lemma:

- $f_k(x)$ represents the $k$-th binary classifier in ECOC, with input $x$ and output in $\{-1, +1\}$.

- $y_k \in \{-1, +1\}$ represents the true label of the $k$-th binary classification problem. $\tilde{y}_k \in \{-1, +1\}$ represents the noisy label.

- The function $f_k$ satisfies the margin condition, i.e., there exist constants $\mu_{1,k}, \mu_{-1,k} \in \mathbb{R}$ and $\gamma_k \in (0,1)$ such that:

$$\mathbb{E}[f_k(x) \mid y_k = 1] \geq \mu_{1,k} \geq 0, \tag{24}$$
$$\mathbb{E}[f_k(x) \mid y_k = -1] \leq \mu_{-1,k} \leq 0, \tag{25}$$
$$\gamma_k = \min\{\mu_{1,k}, -\mu_{-1,k}\}. \tag{26}$$

- The label noise follows the random noise model, i.e., for any $x$, its true label $y_k$ is flipped to $\tilde{y}_k = -y_k$ with probability $\epsilon_k$, independently.

- The activation function $\sigma$ of the network satisfies the $B$-Lipschitz condition, i.e., for any $u, v \in \mathbb{R}$, we have:

$$|\sigma(u) - \sigma(v)| \leq B|u - v|. \tag{27}$$

Let $f_k(x)$ be the output of the $k$-th binary classifier in the ECOC-based DNN, satisfying the assumptions in the lemma. Our goal is to bound the error probability $p_k$ of the binary classifier $f_k$ under the noisy labels $\tilde{y}_k$. First, we consider the case when the true label is $y_k = 1$. By the margin condition, we have:

$$\mathbb{E}[f_k(x) \mid y_k = 1] \geq \mu_{1,k}. \tag{28}$$

Let $\tilde{f}_k(x)$ be the noisy version of $f_k(x)$ under the label noise model, i.e., $\tilde{f}_k(x) = f_k(x)$ with probability $1 - \epsilon_k$ and $\tilde{f}_k(x) = -f_k(x)$ with probability $\epsilon_k$. Then, we have:

$$\mathbb{E}[(\tilde{f}_k(x) - f_k(x))^2 \mid y_k = 1] = 4\epsilon_k \mathbb{E}[f_k^2(x) \mid y_k = 1]. \tag{29}$$

By the Hidden Layer Output Bound (Lemma A.6) and the fact that $n_L = 1$, we have:

$$|f_k(x)| \leq \left(1 - \frac{B + |\phi(0)|}{1 - B}\right) B^L + \frac{B + |\phi(0)|}{1 - B} + \delta$$
$$= \kappa(B, L, \phi(0)) + \delta, \tag{30}$$

with probability at least $1 - \delta$ for any $\delta > 0$. Therefore,

$$\mathbb{E}[f_k^2(x) \mid y_k = 1] \leq (\kappa(B, L, \phi(0)) + \delta)^2. \tag{31}$$

Combining the above inequalities, we get:

$$\mathbb{E}[(\tilde{f}_k(x) - f_k(x))^2 \mid y_k = 1] \leq 4\epsilon_k(\kappa(B, L, \phi(0)) + \delta)^2. \tag{32}$$

Now, applying the Concentration of Noisy Functions (Lemma A.7) with $\sigma^2 = 4\epsilon_k(\kappa(B, L, \phi(0)) + \delta)^2$, we have:

$$\mathbb{P}\left(\tilde{f}_k(x) \leq \mu_{1,k} - 2\sqrt{2\epsilon_k(\kappa(B, L, \phi(0)) + \delta)^2 \log(2/\delta)} \mid y_k = 1\right) \leq \delta \tag{33}$$

Setting $2\sqrt{2\epsilon_k(\kappa(B, L, \phi(0)) + \delta)^2 \log(2/\delta)} = \mu_{1,k}$ and solving for $\delta$, we obtain:

$$\mathbb{P}\left(\tilde{f}_k(x) \leq 0 \mid y_k = 1\right) \leq 2\exp\left(-\frac{\mu_{1,k}^2}{8\epsilon_k(\kappa(B, L, \phi(0)) + \delta)^2}\right)$$
$$\leq 2\exp\left(-\frac{\gamma_k^2}{8\epsilon_k(\kappa(B, L, \phi(0)) + \delta)^2}\right), \tag{34}$$

with probability at least $1 - \delta$. Similarly, for the case when the true label is $y_k = -1$, we can show that:

$$\mathbb{P}\left(\tilde{f}_k(x) \geq 0 \mid y_k = -1\right) \leq 2\exp\left(-\frac{\gamma_k^2}{8\epsilon_k(\kappa(B, L, \phi(0)) + \delta)^2}\right), \tag{35}$$

with probability at least $1 - \delta$. Combining the two cases, we obtain:

$$p_k = \mathbb{P}\left(\tilde{f}_k(x) \leq 0 \mid y_k = 1\right)\mathbb{P}(y_k = 1) + \mathbb{P}\left(\tilde{f}_k(x) \geq 0 \mid y_k = -1\right)\mathbb{P}(y_k = -1)$$
$$\leq 2\exp\left(-\frac{\gamma_k^2}{8\epsilon_k(\kappa(B, L, \phi(0)) + \delta)^2}\right), \tag{36}$$

with probability at least $1 - 2\delta$. Then, with probability at least $1 - \delta$, we have

$$p_k \leq 2\exp\left(-\frac{\gamma_k^2}{8\epsilon_k(\kappa(B,L,\phi(0)) + \delta/2)^2}\right). \tag{37}$$

$\square$

***Proof of Theorem 4.2.*** In Pseudo-Label Learning, suppose the class labels are corrupted by random noise with probability $\epsilon \in (0,1)$, i.e., each true label $y \in [C]$ is flipped to a uniformly random class $\tilde{y} \in [C] \setminus y$ with probability $\epsilon$. Under this noise model, the ECOC-based DNN has a label noise probability of $\epsilon_k = \epsilon \cdot (n+d)/(2n)$ for each binary classifier $f_k$.

**For the ECOC-based DNN,** let $\hat{y} \in [C]$ be the predicted class label for an input $x$, and let $\hat{E}(x) \in \{-1, +1\}^n$ be the corresponding predicted codeword, i.e., $\hat{E}(x)_k = \text{sign}(f_k(x))$ for $k \in [n]$. Let $y \in [C]$ be the true class label of $x$, and let $E([y]) \in \{-1, +1\}^n$ be the corresponding true codeword. Under the given noise model, when the probability that $y$ is flipped to a specific class $\tilde{y} \neq y$, the Hamming distance between $E([y])$ and $E([\tilde{y}])$ is at least $d$, by the definition of the minimum distance of the ECOC matrix. Therefore, for $\hat{y}$ to be misclassified as $\tilde{y}$, the predicted codeword $\hat{E}(x)$ needs to be closer to $E([\tilde{y}])$ than to $E([y])$ in Hamming distance, which means that $\hat{E}(x)$ must differ from $E([y])$ in at least $d/2$ bits. By the union bound, the probability of this event is at most $\binom{n}{d/2}p^{d/2}$, where $p$ is an upper bound on the error probability of each binary classifier. The total probability of misclassification is bounded by

$$\mathbb{P}_{ECOC}(\hat{y} \neq y) \leq \binom{n}{d/2}p^{d/2}. \tag{38}$$

To bound the error probability of each binary classifier, we apply the simplified bound from Lemma A.8 with the label noise probability $\epsilon_k = \epsilon \cdot (n+d)/(2n)$ and probability at least $1 - \delta/n$:

$$p \leq 2\exp\left(-\frac{n\gamma^2}{4(n+d)\epsilon(\kappa(B,L,\phi(0)) + \delta/2n)^2}\right), \tag{39}$$

where $\gamma = \min\{\gamma_k\}$. Finally, the error probability of the ECOC-based classifier is bounded by:

$$\begin{aligned}\mathbb{P}_{ECOC}(\hat{y} \neq y) &\leq \binom{n}{d/2}p^{d/2} \\ &\leq (2en/d)^{d/2}p^{d/2}\end{aligned} \tag{40}$$

with probability at least $1 - \delta$ and using the binomial coefficient bound $\binom{n}{d} \leq (en/d)^d$.

**For the one-hot-based DNN,** the multiclass classifier $f(x)$ predicts the class label of $x$ by taking the argmax of the predicted probabilities:

$$\hat{y} = \arg\max_{j \in [C]} f_j(x). \tag{41}$$

Assume each binary classifier $f_j(x)$ satisfies the margin condition similar as Lemma A.8. Let $\epsilon_j$ be the label noise probability for the $j$-th binary classifier, we have $\epsilon_j = \epsilon$. Then, with probability at least $1 - \delta/C$, the error probability of each binary classifier $f_j(x)$ under noisy labels is bounded by:

$$p_j \leq 2\exp\left(-\frac{\hat{\gamma}_j^2}{8\epsilon(\kappa(B,L,\phi(0)) + \delta/2C)^2}\right). \tag{42}$$

Now, let $E$ be the event that the multiclass classifier $f(x)$ predicts the correct class label under noisy labels, i.e., $\hat{y} = \arg\max_{j \in [C]} y_j$. For event $E$ to hold, it suffices to have $f_j(x) > f_k(x)$ for all $k \neq j$, where $j = \arg\max_{j \in [C]} y_j$ is the true class label. Consider any class $k \neq j$. Under the true labels $y$, we have $y_j = 1$ and $y_k = -1$. By the margin condition, with probability at least $1 - \delta/C$, we have $f_j(x) > f_k(x)$ holds with probability at least $1 - \delta/C$ if:

$$\mu_{j,1} - \hat{\gamma}_j > \mu_{k,-1} + \hat{\gamma}_k. \tag{43}$$

By the union bound, $f_j(x) > f_k(x)$ holds for all $k \neq j$ simultaneously with probability at least $1 - \delta$, implying that event $E$ holds with probability at least $1 - \delta$.

Based on ***Theorem 4.1***, one-hot can be viewed as using $C$ binary classifiers. For correct prediction, the ground-truth score $f_j(x)$ must exceed all others $f_k(x)$ for $k \neq j$, we have the following proof sketch:

1. Define the correct classification event: $E = \bigcap_{k \neq j}\{f_j(x) > f_k(x)\}$

2. The misclassification corresponds to the complement event: $\mathbb{P}(E^{\mathcal{C}}) = \mathbb{P}\left(\bigcup_{k \neq j}\{f_k(x) > f_j(x)\}\right)$

3. Applying the union bound: $\mathbb{P}(E^{\mathcal{C}}) \leq \sum_{k \neq j} \mathbb{P}(f_k(x) > f_j(x))$

4. Using the per-class binary upper bound $p_j$, we obtain the overall one-hot upper bound: $\mathbb{P}_{\text{one-hot}}(\hat{y} \neq y) \leq \sum_{j=1}^{C} p_j$

Finally, the error probability of the multiclass classifier $f(x)$ is bounded by:

$$
\begin{aligned}
\mathbb{P}_{one-hot}(\hat{y} \neq y) = \mathbb{P}(E^{\mathcal{C}}) \\
\leq \sum_{j=1}^{C} p_j \\
\leq 2C \exp\left(-\frac{\hat{\gamma}^2}{8\epsilon(\kappa(B, L, \phi(0)) + \delta/2C)^2}\right),
\end{aligned}
\tag{44}
$$

with probability at least $1 - \delta$, where $\hat{\gamma} = \min\{\hat{\gamma}_j\}$.

To derive the condition on the minimum distance $d$ for the ECOC-based DNN to have a tighter error bound than the one-hot encoding DNN, we compare the two bounds (Equations 40 and 44):

$$
(\frac{2en}{d})^{\frac{d}{2}} \exp\left(-\frac{nd\gamma^2}{8(n+d)\epsilon(\kappa(B, L, \phi(0)) + \delta/2n)^2}\right) < 2C \exp\left(-\frac{\hat{\gamma}^2}{8\epsilon(\kappa(B, L, \phi(0)) + \delta/2C)^2}\right)
$$

$$
\left(\frac{2en}{d}\right)^{\frac{d}{2}} \exp\left(-\frac{nd\gamma^2}{8(n+d)\epsilon(\kappa + \delta/2n)^2}\right) < 2C \exp\left(-\frac{\hat{\gamma}^2}{8\epsilon(\kappa + \delta/2C)^2}\right)
$$

$$
\Rightarrow \quad \left(\frac{2en}{d}\right)^{\frac{d}{2}} \exp\left(-\frac{nd\gamma^2}{8(n+d)\epsilon\kappa^2}\right) < 2C \exp\left(-\frac{\hat{\gamma}^2}{8\epsilon\kappa^2}\right)
$$

$$
\Rightarrow \quad \left(\frac{2en}{d}\right)^{\frac{d}{2}} \exp\left(-\frac{nd\gamma^2}{8(n+d)\epsilon\kappa^2}\right) < 2C \exp\left(-\frac{\hat{\gamma}^2}{8\epsilon\kappa^2}\right)
$$

$$
\Rightarrow \quad \frac{d}{2}\log\left(\frac{2en}{d}\right) - \frac{nd\gamma^2}{8(n+d)\epsilon\kappa^2} < \log(2C) - \frac{\hat{\gamma}^2}{8\epsilon\kappa^2}
$$

$$
\Rightarrow \quad \frac{nd\gamma^2}{8(n+d)\epsilon\kappa^2} - \frac{\hat{\gamma}^2}{8\epsilon\kappa^2} > \frac{d}{2}\log\left(\frac{2en}{d}\right) - \log(2C)
$$

$$
d > \frac{8(n+d)\epsilon\kappa^2}{n\gamma^2}\left(\frac{d}{2}\log\left(\frac{4en}{d}\right) - \log(2C) + \frac{\hat{\gamma}^2}{8\epsilon\kappa^2}\right)
\tag{45}
$$

Using the fact that $d < n$, and the monotonicity of $\frac{lnx}{x}$, we have:

$$
d > \frac{16\epsilon\kappa^2}{\gamma^2}\left(\frac{(1+log2)n}{2} - \log(2C)\right) + 2\frac{\hat{\gamma}^2}{\gamma^2}
\tag{46}
$$

The equation suggests that when the noise level ($\epsilon$) is higher, we can use a larger code distance ($d$) to obtain a tighter bound for ECOC-based classifer. The lower bound of $d$ is determined by the ratio of the margins of the two classifiers ($2\frac{\hat{\gamma}^2}{\gamma^2}$). $\qquad \square$

**Remark on Assumptions.** The theoretical results presented above rely on two standard assumptions: *(i)* classifier independence and *(ii)* uniform random noise. Both assumptions are analytically tractable and justified in the context of our analysis.

**(1) Independence Assumption.**

- Theorem 4.1 analyzes ECOC decoding from a global perspective and does **not require** classifier independence.

- Theorem 4.2 assumes bit-wise independence only to derive a **worst-case error bound**. In practice, classifier correlations tend to **reduce** joint error, so the assumption does **not compromise** the theorem's validity. Specifically,

$$\mathbb{P}_{\text{ECOC}}(\hat{y} \neq y) \leq \left(\frac{2en}{d}\right)^{d/2} p^{d/2}.$$

When classifiers are correlated, the expected joint error is lower than the product of marginal probabilities due to Jensen's inequality:

$$\mathbb{E}[p^{d/2}] \leq (\mathbb{E}[p])^{d/2}.$$

Therefore, modeling such dependencies would **tighten** the bound, and the assumption of independence yields a **conservative** estimate.

In practice, perfect independence is rarely satisfied. Even with orthogonal or class-agnostic codes (e.g., one-hot or $M_{\text{mmd}}$), optimization often introduces correlations. Explicitly modeling these dependencies is non-trivial and beyond our scope. Thus, the independence assumption offers a **tractable and analyzable abstraction**.

**(2) Uniform Noise Assumption.**

We adopt uniform random noise as a clean baseline, following standard theoretical practice. While real-world pseudo-label noise may exhibit correlations and structured patterns, this assumption does **not undermine** our results:

- At the class level, such structure affects both one-hot and ECOC **similarly**, as both operate at the pixel level. Our primary goal is to replace one-hot encoding with ECOC within pseudo-label learning frameworks, and thus the relative comparison remains valid. Furthermore, our Reliable Bit Mining mechanism partially mitigates such structure by exploiting semantic relationships across classes.

- At the bit level, correlations make the independence-based ECOC bound more **conservative** (see **Independence Assumption**), since correlations typically reduce joint error. As a result, practical performance often surpasses the worst-case bound derived under independence.

## B  Implementation of Encoding Strategy

In this section, we give implementation details of two algorithms for codebook generation, as shown in Alg. 2 and Alg. 3, respectively.

---

**Algorithm 2** Max-min distance encoding

---

**Input:** classes number $N$, codeword length $K$, iterations $L$
**Output:** binary matrix of codebook $M_{mmd} \in \{0,1\}^{N \times K}$
1: $D_{sum} = 0$
2: **for** $j = 1$ **to** $L$ **do**
3:      Generate random binary matrix $m \in \{0,1\}^{N \times K}$
4:      Compute $d_{min\_r}, d_{min\_c}, d_{max\_c}$
5:      **if** $d_{min\_r} = 0$ **or** $d_{min\_c} = 0$ **or** $d_{max\_c} = N$ **then**
6:          **continue**
7:      **end if**
8:      $d_{sum} = d_{min\_r} + d_{min\_c} + N - d_{max\_c}$
9:      **if** $d_{sum} > D_{sum}$ **then**
10:         $D_{sum} = d_{sum}, M_{mmd} = m$
11:      **end if**
12: **end for**
13: **return** $M_{mmd}$

---

**Max-min distance encoding.** A good error-correcting output code should satisfy **row separation** and **column separation**. For the former, the minimum Hamming distance between each pair of codewords $d_{min\_r}$ should be maximized, which can correct at least $\lfloor \frac{d_{min\_r}-1}{2} \rfloor$ single bit errors. As for the latter, each bit classifier should be uncorrelated with the others. This can be ensured by maximizing $d_{min\_c}$ and $N - d_{max\_c}$, which are calculated between columns of the code matrix. Although searching for an optimal encoding matrix is known as an NP-hard problem [20], we can obtain a sufficiently valuable encoding matrix, denoted as $M_{mmd}$, through a random generation strategy due to the sparsity of the encoding space.

Due to the large size of the encoding space ($2^K$) relative to the number of classes $N$, a randomly generated codebook typically ensures a sufficiently large minimum Hamming distance. Any pair of such random strings will be separated by a Hamming distance that is binomially distributed with mean $K/2$. We can search for the appropriate codebook through multiple iterations to satisfy the optimal row separation and column separation. At the same time, we should ensure the validity of the codebook from two perspectives: (1) There are no identical codewords, which can be ensured by $d_{min\_r} > 0$. (2) There are no wholly identical or opposite classifiers, which can be ensured by $d_{min\_c} > 0$ and $d_{max\_c} < N$. In our case, we set iterations $L = 100000$ to generate a sufficiently robust codebook.

---

**Algorithm 3** Text-based encoding

---

**Input:** classes names $\{class\}$, codeword length $K$,
**Output:** binary matrix of codebook $M_{text} \in \{0,1\}^{N \times K}$

1: "$\{class\}$" $\xrightarrow{word2vec}$ $f_{text} \in \mathbb{R}^{N \times C}$
2: Scale by the L2 norm $\bar{f}_{text} = f_{text}/\|f_{text}\|_2$
3: Calculate channel-wise variance $\sigma \in \mathbb{R}^C$
4: Sort $\bar{f}_{text}$ in descending order based on $\sigma$
5: $k = 1$
6: **for** $\bar{f} \in \mathbb{R}^N$ **in** $\bar{f}_{text} \in \mathbb{R}^{N \times C}$ **do**
7:     Calculate the mean $m$ of $\bar{f}$
8:     **for** $n = 1$ **to** $N$ **do**
9:       **if** $\bar{f}_n < mean$ **then**
10:         $M_{text}[n,k] = 0$
11:       **else**
12:         $M_{text}[n,k] = 1$
13:       **end if**
14:     **end for**
15:     **if** $M_{text}$ $is$ $valid$ **then**
16:       $k = k + 1$
17:       **if** $k > K$ **then**
18:         **break**
19:       **end if**
20:     **end if**
21: **end for**
22: **return** $M_{text}$

---

**Text-based encoding.** According to the above criteria, we can design encoding matrices that exhibit desirable properties and sufficient robustness. However, the resulting codewords are class-agnostic, and the corresponding binary classification problem may be difficult to optimize. Other than manually designing encodings based on class attributes, we adopt a concise and automated method based on text embedding to generate codewords for the classes. Specifically, $N$ class names are mapped to the feature space $f_{text} \in \mathbb{R}^{N \times C}$ through word2vec [60]. Then, we compress extracted continuous features and quantize them into binary encodings of length $K$ to obtain the code matrix, denoted as $M_{text}$. This encoding strategy considers the relationships and structural information among classes, facilitating more efficient encoding learning.

To consider the relationships and structural information among classes, we resort to word2vec [60] to extract class-related features. Then, we select the most discriminant feature components based on variance magnitude to compress the feature dimensionality to the length of the codeword. Furthermore, we quantize the features into 0-1 encoding using the mean of the feature components as

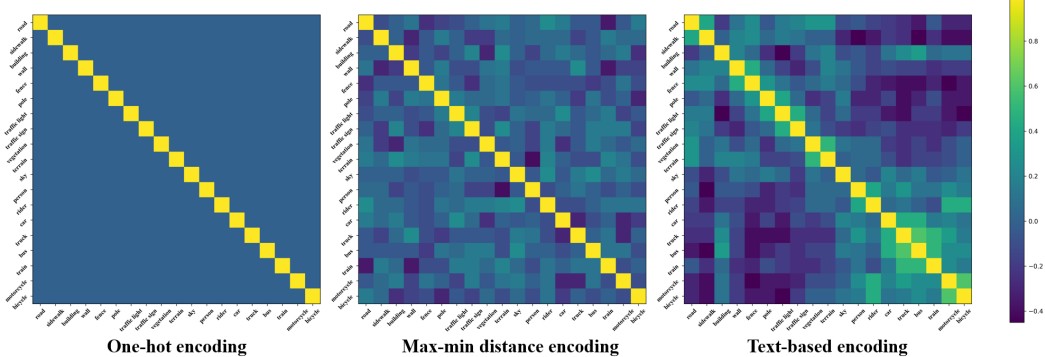

Figure 5: The similarity matrix of different encoding forms for 19 classes in Cityscapes, where the minimum code distance in $M_{mmd}$ is 15 and in $M_{text}$ is 8. Note that we standardize the binary value $\{0, 1\}$ to $\{-1, 1\}$ to calculate similarity for ECOC encoding.

a threshold. During this process, we also need to ensure the validity of the codebook, as discussed above.

## C  Analysis of Coding Strategy

We first visualize the similarity matrix of different encoding forms in Fig. 5. The classes are typically encoded in a one-hot form, which is easily susceptible to the influence of label drift in the pseudo-label learning process. The $M_{mmd}$ aims to maximize the distance between classes, ensuring the robustness of the labels. Furthermore, $M_{text}$ takes into account the relationships and structures between different classes, ensuring that similar classes have similar encodings. This property makes the resulting binary classification problems easier to optimize.

Due to the larger code distance in $M_{mmd}$, it may generate more label noise for the code-wise form of pseudo-labels, leading to erroneous training in pseudo-label learning and limiting performance improvement. However, because of its sufficient robustness, bit-wise pseudo-labels can provide more stable performance gains for $M_{mmd}$. In $M_{text}$, this phenomenon is reversed because $M_{text}$ naturally perceives the relationships between classes, making its encoding form easy to learn, and thus code-wise pseudo-labels achieve higher performance, as shown in Table 4. Our hybrid pseudo-labels effectively combine the advantages of both forms, achieving the best performance gains.

To further study the different selections for coding strategy, we conduct experiments built with denoted as $M_{mmd}$ and $M_{text}$ respectively, and evaluate performance on UDA setting with DAFormer [34] and SSL setting with UniMatch [93]. The results are shown in Table 5 and Table 6. We also We present a comprehensive comparison in Figure 6. Each code design has its own benefits:

- $M_{mmd}$ ensures larger code distances and stronger error correction.
- $M_{text}$ preserves semantic relationships between classes, which helps in learning easier-to-optimize binary classifiers.

Both $M_{mmd}$ and $M_{text}$ show consistent competitive performance, meaning that the pseudo-label learning process can benefit from ECOC encoding and proposed hybrid pseudo-labels. In our experiments, we implement $M_{text}$ as the default setting for ECOCSeg.

Table 5: Results on Cityscapes of **UDA setting** built with DAFormer [34].

| Source dataset | baseline | $M_{mmd}$ | $M_{text}$ |
|---|---|---|---|
| GTAv | 68.3 | 70.2 | **70.5** |
| SYNTHIA | 60.9 | 63.1 | **63.3** |

Note that our encoding strategies, focusing on class separability, visual similarity, and codeword length. These components are essential for building effective codebooks in ECOCSeg.

Table 6: Results on Pascal of **SSL setting** bulit with UniMatch [93].

| Partition protocol | baseline | $M_{mmd}$ | $M_{text}$ |
|---|---|---|---|
| 1/16 | 76.5 | 78.1 | **78.1** |
| 1/4 | 77.2 | 78.8 | **78.9** |

- **Class Separability.** We propose two simple yet effective codebook generation strategies:
  - $M_{mmd}$: A class-agnostic strategy that maximizes the minimum pairwise Hamming distance among codewords.
  - $M_{text}$: A text-guided strategy that ensures code diversity via balanced 0/1 quantization of pretrained language embeddings.

  Both strategies guarantee sufficient inter-codeword Hamming distance, which is crucial for the error-correction capability of ECOC. As visualized in Figure 5, the constructed codebooks provide meaningful inter-class separation in the Hamming space.

- **Visual Similarity.** Incorporating visual similarity into the codebook is optional and not strictly required—similar to how traditional one-hot encoding is class-agnostic by design. Specifically, $M_{mmd}$ does not rely on class semantics, while $M_{text}$ implicitly captures semantic and visual similarity via language priors from pretrained text embeddings. These strategies are compatible and effective under various settings, as demonstrated in Figure 6.

- **Codeword Length.** We analyze the impact of codeword length $K$ in Appendices E and G, including selection criteria and empirical results. Our findings indicate that moderately long codes offer a favorable trade-off between robustness and computational efficiency.

# D    Analysis of Threshold $T$

We analyze the influence of $T$ in Table 4, and further present sensitivity results on different benchmarks in Figure 6. Based on these analyses, we conclude:

- Both code-wise and bit-wise pseudo-labels independently improve the performance due to the explicit attribute-level decoupling enabled by ECOC.

- In the Reliable Bit Mining, $T = 0.5$ and $T = 1$ correspond to the pure code-wise and bit-wise forms respectively, while intermediate values yield more robust hybrid labels. This makes $T$ a non-sensitive hyperparameter.

- The choice of $T$ is robust across datasets. We use a fixed setting of $T = 0.95$ in all experiments, achieving consistent performance gains.

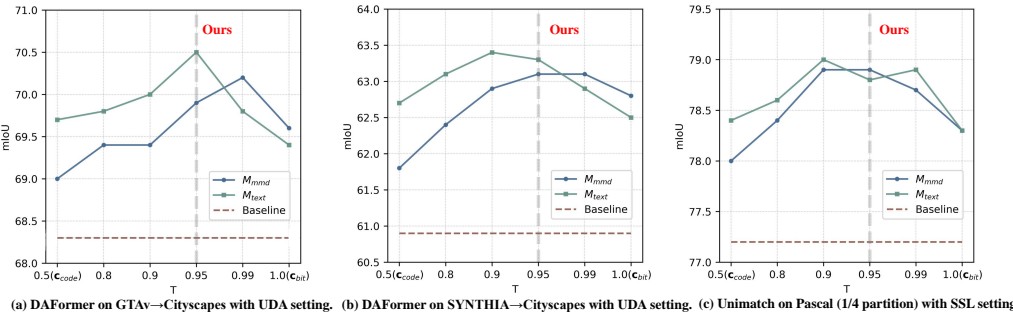

(a) DAFormer on GTAv→Cityscapes with UDA setting.  (b) DAFormer on SYNTHIA→Cityscapes with UDA setting.  (c) Unimatch on Pascal (1/4 partition) with SSL setting.

Figure 6: Sensitivity analysis of $T$ across different benchmarks.

# E    Analysis of Encoding Length $K$

Generally, an adequate length of codewords (at least $\log N$) is required to ensure robust encoding. However, excessively long codewords can lead to redundancy and inefficient optimization. Table

7 studies the influence of encoding length $K$. ECOCSeg performs well even with a low encoding length of $K = 10$, which is lower than the number of classes $N = 19$. The performance improves continuously as the encoding length increases within the range of less than 40. However, when $K > 40$, the performance shows negligible improvement. To balance performance and computational costs, we select $K = 40$ for both Cityscapes (19 classes) and Pascal (21 classes). In Appendix G, we also show that ECOCSeg can handle a larger number of classes efficiently.

Table 7: Ablation study on $K$, built with DAFormer under the **fully supervised learning setting** on Cityscapes.

| $K$ | 10 | 20 | 30 | 40 | 50 | 60 |
|------|------|------|------|------|------|------|
| mIoU | 77.1 | 77.5 | 78.0 | 78.1 | 78.1 | 78.2 |

## F    Analysis of Shared Bits

The assumption used in Reliable Bit Mining (sec. 3.4) that "shared bits are more reliable" stems from observation that "correct class is often among Top-$C$ nearest neighbors" in the codeword space. This is a widely observed property, as confusing classes often lie close to each other in the feature or code space due to small discriminative margins.

This is empirically confirmed in Figure 7:

- Across benchmarks, mAcc drops monotonically with lower-ranked predicted classes.
- Aggregated mAcc over Top-$C$ classes quickly approaches 1, indicating correct class is usually included.
- Since correct class's codeword contains only correct bits, the shared bits from Top-$C$ set are guaranteed to be accurate when correct class is included.

Therefore, our Reliable Bit Mining strategy leverages this phenomenon to robustly extract accurate bits, even in the presence of pixel-level noise.

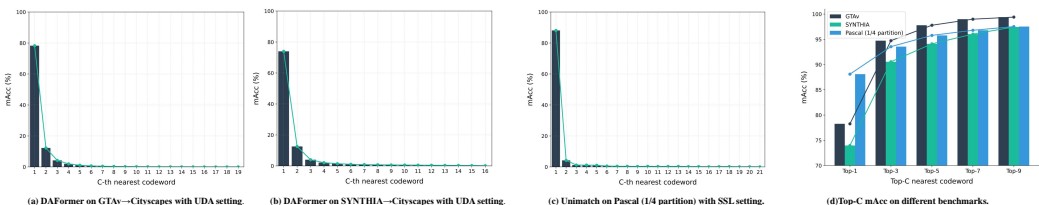

Figure 7: (a,b,c) Histogram of mAcc corresponding to the C-th nearest codeword. (d) Accumulated mAcc of the Top-C classes on different benchmarks.

## G    More Experiments on SSL

In this section, we provide more experiment settings and results on SSL on more powerful baselines, challenging benchmarks and real-world scenarios. We emphasize that ECOCSeg is designed as a label representation improvement, independent of network architecture or training strategy. Therefore, it can be seamlessly integrated into a wide range of semi-supervised learning (SSL) frameworks as a plug-and-play module.

**Challenging Benchmarks.** Follow the basic setting of UniMatch [93], the initial learning rate is set as 0.005 and 0.004 for Cityscapes and COCO respectively, with a SGD optimizer. The model is trained for 240, and 30 epochs under a poly learning rate scheduler. The training resolution is set as 801, and 513 for these two datasets. We adopt the Xception-65 [18] as backbone when trained on COCO. In Table 8, we implement the experiments on Cityscapes. ECOCSeg consistently outperforms baselines with gains ranging from 1.0% to 3.2%, especially on most challenging 1/16 partition, verifying its robustness and generalization ability. We also show the results on COCO in Table 9, where ECOCSeg outperforms the baseline with gains ranging from 1.7% to 2.6%. We find ECOC is

efficient to handle a larger number of classes due to its binary decoupling property. Theoretically, a length of $K = \log_2 N$ is sufficient to represent classes, while excessively long encodings can lead to redundancy. Empirically, the optimal code length is roughly $K = 10 \log_2 N$ [2]. Based on this empirical rule, we use $K = 60$ for COCO (81 classes) and conduct experiments with UniMatch, observing significant improvements.

Table 8: SSL segmentation performance on Cityscapes. The 321 and 513 denote the training resolution.

| Method | ResNet-50 | | | | ResNet-101 | | | |
|---|---|---|---|---|---|---|---|---|
| | 1/16 | 1/8 | 1/4 | 1/2 | 1/16 | 1/8 | 1/4 | 1/2 |
| Sup-only | 63.3 | 70.2 | 73.1 | 76.6 | 66.3 | 72.8 | 75.0 | 78.0 |
| U$^2$PL [87] | 70.6 | 73.0 | 76.3 | 77.2 | 74.9 | 76.5 | 78.5 | 79.1 |
| AugSeg [102] | 73.7 | 76.4 | 78.7 | 79.3 | 75.2 | 77.8 | 79.5 | 80.4 |
| DAW [77] | 75.2 | 77.5 | 79.1 | 79.5 | 76.6 | 78.4 | 79.8 | 80.6 |
| RankMatch [58] | 75.4 | 77.7 | 79.2 | 79.5 | 77.1 | 78.6 | 80.0 | 80.7 |
| Fixmatch [72] | 72.6 | 75.7 | 76.8 | 78.2 | 74.2 | 76.2 | 77.2 | 78.4 |
| **+ECOCSeg** | **75.8** | **78.1** | **78.5** | **79.3** | **77.3** | **78.4** | **78.9** | **79.4** |
| UniMatch [93] | 75.0 | 76.8 | 77.5 | 78.6 | 76.6 | 77.9 | 79.2 | 79.5 |
| **+ECOCSeg** | **77.1** | **78.2** | **78.9** | **79.6** | **78.2** | **79.3** | **80.5** | **80.7** |

Table 9: SSL segmentation performance on COCO.

| Method | 1/512 | 1/256 | 1/128 | 1/64 | 1/32 |
|---|---|---|---|---|---|
| Sup-only | 22.9 | 28.0 | 33.6 | 37.8 | 42.2 |
| PseudoSeg [106] | 29.8 | 37.1 | 39.1 | 41.8 | 43.6 |
| PC$^2$Seg [104] | 29.9 | 37.5 | 40.1 | 43.7 | 46.1 |
| AllSpark [82] | 34.1 | 41.6 | 45.4 | 49.5 | 50.9 |
| UniMatch [93] | 31.9 | 38.9 | 44.4 | 48.2 | 49.8 |
| **+ECOCSeg** | **34.5** | **41.8** | **46.2** | **49.9** | **51.6** |

**Powerful Baselines.** To demonstrate the compatibility of our method with modern architectures and training paradigms, we conduct additional experiments on UniMatch V2 [94] using the Pascal VOC high-quality set, with a DINOv2-S [63] encoder. The results are summarized in Table 10. These results confirm that ECOCSeg consistently improves performance, even when built upon strong semi-supervised learning baselines with large-scale pre-training.

Table 10: SSL segmentation performance on Pascal VOC (high-quality set).

| Setting | 1/16 (92) | 1/8 (183) | 1/4 (366) | 1/2 (732) | Full (1464) |
|---|---|---|---|---|---|
| AugSeg [102] (RN-101) | 71.1 | 75.5 | 78.8 | 80.3 | 81.4 |
| CorrMatch [73] (RN-101) | 76.4 | 78.5 | 79.4 | 80.6 | 81.8 |
| BeyondPixels [33] (RN-101) | 77.3 | 78.6 | 79.8 | 80.8 | 81.7 |
| UniMatch V2 [94] (DINOv2-S) | 79.0 | 85.5 | 85.9 | 86.7 | 87.8 |
| + ECOCSeg | **81.1** | **86.6** | **87.1** | **87.8** | **88.9** |

**Real-World Scenarios.** To evaluate the generalizability of ECOCSeg beyond natural scene segmentation, we conduct experiments on two real-world, label-scarce domains: *remote sensing* and *medical imaging*.

*Remote Sensing:* We integrate ECOCSeg into UniMatch [93] (PSPNet [101]) for binary change detection on the WHU-CD dataset [39]. To thoroughly evaluate the effectiveness of ECOCSeg, we split the WHU-CD dataset into three subsets following previous methods [93]: a training set containing 5,947 images,a verification setwith 743 images,and a test set comprising 744 images. The results are shown in Table 11.

Note that WHU-CD is a binary classification task, where ECOC encoding is not meaningful due to the absence of class diversity. In this case, the only difference introduced by ECOCSeg lies in the quality estimation strategy:

Table 11: Binary change detection results on WHU-CD (PSPNet).

| Method | 5% | 10% | 20% | 40% |
|---|---|---|---|---|
| Sup-only | 48.3 | 60.7 | 69.7 | 69.5 |
| S4GAN [61] | 18.3 | 62.2 | 70.8 | 76.4 |
| SemiCDNet [65] | 51.7 | 62.0 | 66.7 | 75.9 |
| SemiCD [4] | 65.8 | 68.1 | 74.8 | 77.2 |
| UniMatch [93] | 77.5 | 78.9 | 82.9 | 84.4 |
| + ECOCSeg | **78.0** | **79.6** | **83.5** | **84.6** |

- **UniMatch** uses threshold-based filtering;
- **ECOCSeg** adopts a global confidence-based quality score (see Appendix H).

We find that this modification leads to a modest performance gain.

*Medical Imaging:* We also evaluate ECOCSeg on the ACDC dataset [5], a four-class cardiac MRI segmentation task. Results using UniMatch with a UNet [68] backbone are shown in Table 12.

Table 12: Multi-class segmentation results on ACDC (UNet).

| Method | 1 Case | 3 Cases | 7 Cases |
|---|---|---|---|
| Sup-only | 28.5 | 41.5 | 62.5 |
| UA-MT [97] | - | 61.0 | 81.5 |
| CPS [12] | - | 60.3 | 83.3 |
| CNN&Trans [57] | - | 65.6 | 86.4 |
| UniMatch [93] | 85.4 | 88.9 | 89.9 |
| + ECOCSeg | **86.7** | **90.1** | **90.5** |

Unlike WHU-CD, ACDC benefits more substantially from ECOCSeg due to the presence of multiple correlated classes. This enables our bit-level label refinement mechanism to take effect, improving segmentation quality.

These experiments demonstrate that ECOCSeg is applicable across diverse real-world domains. However, as discussed above, the advantages of ECOCSeg are more **pronounced in multi-class settings**, where fine-grained class disentanglement and bit-level shared attributes play a larger role.

## H   Analysis of Quality Estimate

In pseudo-label learning, we need a quality estimate $q(p_{ij})$ to control the optimization process for $\mathcal{L}^u$, where $p_{ij}$ is confidence and typically defined by maximum class probability. In DAFormer [34], it is defined by:

$$q^{(i)} = \frac{\sum_{j=1}^{H \times W} [p_{ij} > \tau']}{H \times W}, \tag{47}$$

where $[\cdot]$ denotes the Iverson bracket, and this weight is applied to all pixels of the entire image simultaneously. In UniMatch [93], it is defined through a threshold filtering way:

$$q(p_{ij}) = [p_{ij} > \tau'], \tag{48}$$

meaning that only high-confidence pixel samples are used for training. To study the differences between the two estimate methods, we implement ablation experiments based on UniMatch, using 1/4 labeled data with ResNet-101 on the Pascal dataset, as shown in Table 13. We set the same threshold $\tau' = 0.95$ to ensure similar proportions of loss introduced by pseudo-labels. For Eq. 48, due to the selection of high-confidence samples only, the robustness of the pseudo label generated by ECOCSeg is meaningless, and the introduced reliable bit mining mechanism does not work in this

way, leading to no advantage compared to the one-hot encoding approach. For Eq. 47, the training process is modulated by gradually increasing weights, and all samples are taken into consideration. In this way, the one-hot encoding approach faces a performance decline due to the introduction of a large number of erroneous labels, while ECOCSeg can handle this label noise and benefit from the sufficient training of all samples, resulting in a significant performance improvement. Based on the above discussions, we implement Eq. 47 for ECOCSeg and set the same threshold $\tau'$ as corresponding baseline methods to evaluate the performance.

Table 13: The performances based on different quality estimates on Pascal of **SSL setting**.

| Method | Eq. 47 | Eq. 48 |
|---|---|---|
| UniMatch | 76.7 | 77.2 |
| +ECOCSeg | **78.9** | 77.0 |

Table 14: Analysis of hyper-parameter in optimization criterion, built with DAFormer under the **fully supervised learning setting**.

| $\lambda_1$ | 0.5 | 2 | 5 | 10 | 20 | 50 |
|---|---|---|---|---|---|---|
| mIoU | 77.9 | **78.1** | **78.1** | **78.1** | 78.0 | 77.8 |
| $\lambda_2$ | 0.1 | 0.5 | 1 | 2 | 5 | 20 |
| mIoU | 77.8 | 77.9 | **78.1** | **78.1** | 78.0 | 77.6 |
| $\tau$ | 0.1 | 0.2 | 0.5 | 1 | 2 | 5 |
| mIoU | 77.5 | 77.7 | **78.1** | 78.0 | 77.6 | 77.2 |

## I   Influence of Parameters Setting

In this section, we study the hyper-parameter setting introduced in the optimization criterion for ECOCSeg, i.e., $\lambda_1$, $\lambda_2$, and $\tau$. All experiments are built with DAFormer of a fully supervised learning setting on Cityscapes, and results are summarized in Table 14. We observe that ECOCSeg is robust to the two coefficients, $\lambda_1$ and $\lambda_2$, and achieves the best performance at $\lambda_1 = 5$, $\lambda_2 = 2$. The setting of temperature $\tau$ has a more significant impact, and we set it to 0.5 for stable performance.

## J   ECOCSeg Efficiency Analysis

We provide a detailed comparison of training cost between baseline methods and their ECOCSeg-enhanced counterparts. The results are summarized in Table 15.

Table 15: Training cost comparison between baseline methods and ECOCSeg versions.

| Method | FLOPs (G) | GPU Memory (MB) | Training Time (h) |
|---|---|---|---|
| DAFormer [34] | 116.64 | 9,807 | 13.7 |
| + ECOCSeg | 116.72 | 12,817 | 16.2 |
| UniMatch [93] | 96.16 | 19,542×2 | 16.8 |
| + ECOCSeg | 96.24 | 24,314×2 | 21.5 |

**FLOPs:** ECOCSeg only modifies the final classification head, changing the output dimension from $N$ (number of classes) to $K$ (code length). This introduces a negligible increase in FLOPs. Therefore, the inference cost remains nearly unchanged.

**Memory and Training Time:** The additional memory and training time primarily stem from two components:

- A small ECOC codebook, represented as a binary matrix of size $N \times K$.
- The lightweight Reliable Bit Mining algorithm introduced during training.

These components incur modest overhead, which is acceptable considering the consistent performance improvements observed across various benchmarks.

Importantly, our method operates purely in the label representation space and is **orthogonal** to network architectures or training strategies. As a result, ECOCSeg can be seamlessly integrated into a wide range of pseudo-label learning frameworks as a **plug-and-play** module. This is further validated through experiments across both unsupervised domain adaptation (UDA) and semi-supervised learning (SSL) settings, demonstrating its broad applicability and efficiency.

## K   Confidence Calibration

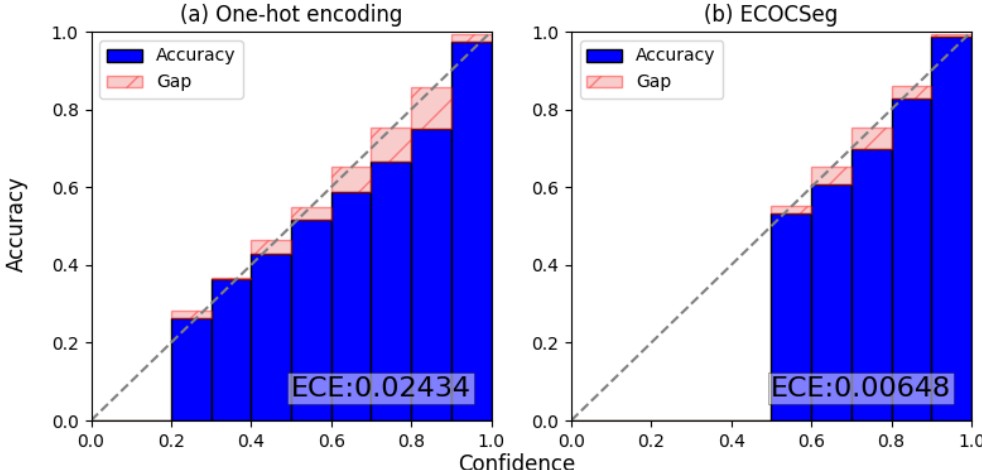

Figure 8: Reliability diagrams for DeepLabv3+ [10] based on one-hot encoding (a) and ECOCSeg (b) on Pascal val.

In this section, we further study the model calibration of ECOCSeg and compare it with the one-hot encoding form. We implement DeepLabv3+ [10] based on these two paradigms and evaluate on Pascal val. In Fig. 8, we present the reliability diagrams, which plot the expected pixel accuracy as a the function of confidence, and calculate the Expected Calibration Error (ECE) [28]:

$$ECE = \sum_{m=1}^{M} \frac{|B_m|}{n} |acc(B_m) - conf(B_m)|, \tag{49}$$

where we divide the bins with an interval of 0.1. Note that the reliability diagram is obtained through the bit-wise way in ECOCSeg, meaning that every pixel will provide $K$ samples in total. As observed, ECOCSeg demonstrates smaller gaps between the expected accuracy and confidence, indicating superior calibration of predictions. While the one-hot encoding form is typically notorious for inflating the probability of the predicted class [70] and suffers higher calibration error accordingly, ECOCSeg exhibits better reliability and interpretability through fine-grained bit-level label representation.

## L   Comparison with Previous Methods

We compare our method with representative approaches that focus on fine-grained modeling and pseudo-label refinement, as summarized in Table 16. While fine-grained representation learning is a common and intuitive strategy, it is not the central contribution of our work. Instead, our method introduces a novel perspective by addressing pseudo-label noise through the lens of the *label encoding space*, rather than the feature space.

ECOCSeg is orthogonal to existing methods that operate primarily at the representation level. By leveraging error-correcting output codes and explicitly modeling inter-class separability through binary attributes, we provide a complementary and scalable solution. We believe this direction opens up new opportunities for robust pseudo-label learning and could inspire further research in this area.

Table 16: Comparison with previous methods

| | Task | Motivation | Implementation | Key Difference |
|---|---|---|---|---|
| [1] [7] | Weakly-supervised | Discriminative regions yield incomplete pseudo-labels | Alternating optimization: 1) cluster sub-classes per class; 2) train a sub-class classifier | Sub-class discovery at the feature level |
| [2] [40] | Domain adaptive | Prior works focus on intra-class alignment without explicitly modeling inter-class structure | Online prototype update for contrastive learning | Structure modeling via class prototypes |
| [3] [100] | Domain adaptive | Improve pseudo-labels via feature clustering | Prototype-based denoising | Feature-space method |
| [4] [71] | Domain adaptive | Avoid manual thresholds | Symmetric distillation & consensus | Threshold-free, feature-level |
| [5] [98] | Domain adaptive | Fuse online-offline pseudo-labels | Unified multi-branch fusion | Fusion in feature space |
| [6] [62] | Active Domain adaptive | Using a small amount of labeled target data to guide adaptation | Anchor-based soft alignment | Feature-space alignment |
| [7] [59] | Semi-supervised | Treat prediction error as a learnable correction term | Introduce a correction network that learns to refine predictions with residual errors. | Prediction refinement via residual correction |
| [8] [24] | Semi-supervised | Reduce confirmation bias | Improve pseudo-labels by peeking at future model states | Modification of the teacher model |
| Ours | Domain adaptive/Semi-supervised | SConfusing classes share attributes — we exploit this to reduce pseudo-label noise | 1) Use ECOC-based classification to decouple classes into shared binary attributes; 2) Design Reliable Bit Mining and hybrid pseudo-labels to denoise | 1) Bit-level denoising via ECOC; 2) Label encoding perspective, orthogonal to prior feature-level methods |

## M    Qualitative Results

In this section, we provide more qualitative results to compare ECOCSeg and corresponding baseline methods on different benchmarks. As shown in Fig. 9 and Fig. 10, the baseline methods face challenges in distinguishing between confusing classes such as *sidewalk* and *road*, *pole* and *building*, *bus* and *truck*, *cow* and *horse*, and so on. These classes are challenging to learn in pseudo-label learning due to the influence of label drift. When built with ECOCSeg, there is a notable improvement in the performance of these classes, which further demonstrates the effectiveness of our method.

## N    Limitation and Impact Statement

While ECOCSeg demonstrates consistent improvements across a wide range of benchmarks, several limitations remain. First, the effectiveness of the ECOC encoding depends on the quality of the codebook design. Although we propose two practical strategies ($M_{mmd}$ and $M_{text}$), sub-optimal

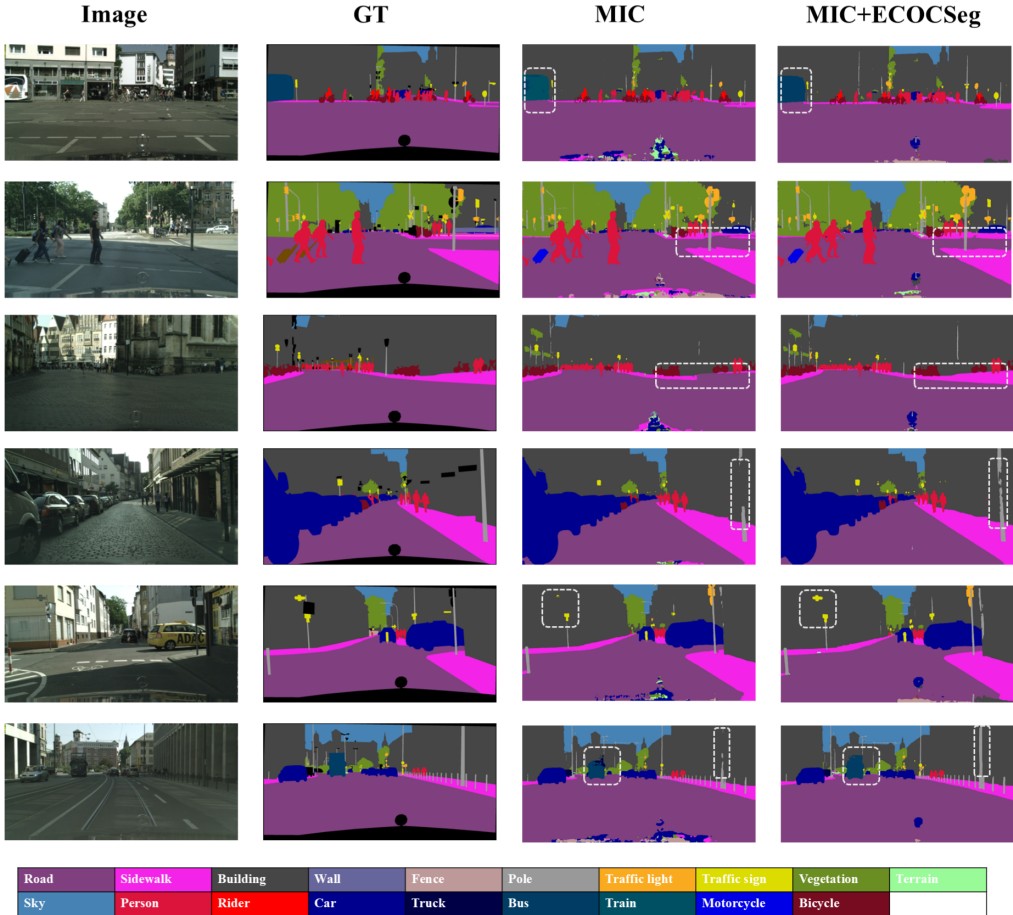

| Image | GT | MIC | MIC+ECOCSeg |

| Road | Sidewalk | Building | Wall | Fence | Pole | Traffic light | Traffic sign | Vegetation | Terrain |
|---|---|---|---|---|---|---|---|---|---|
| Sky | Person | Rider | Car | Truck | Bus | Train | Motorcycle | Bicycle | |

Figure 9: More qualitative comparison built with MIC [35] on UDA benchmark of GTAv→Cityscapes. The significant improvements are marked with dotted boxes.

codeword configurations may still hinder performance in certain edge cases. Second, our theoretical analysis assumes bit-wise independence and uniform label noise, which may not fully capture the structured or correlated noise patterns commonly observed in real-world scenarios. These assumptions, while analytically tractable, may limit the theoretical guarantees when applied to more complex distributions.

Within this paper, we present an approach for pseudo-label learning, especially domain adaptive/semi-supervised semantic segmentation, a pivotal research area in the realm of computer vision, with no apparent negative societal implications known thus far.

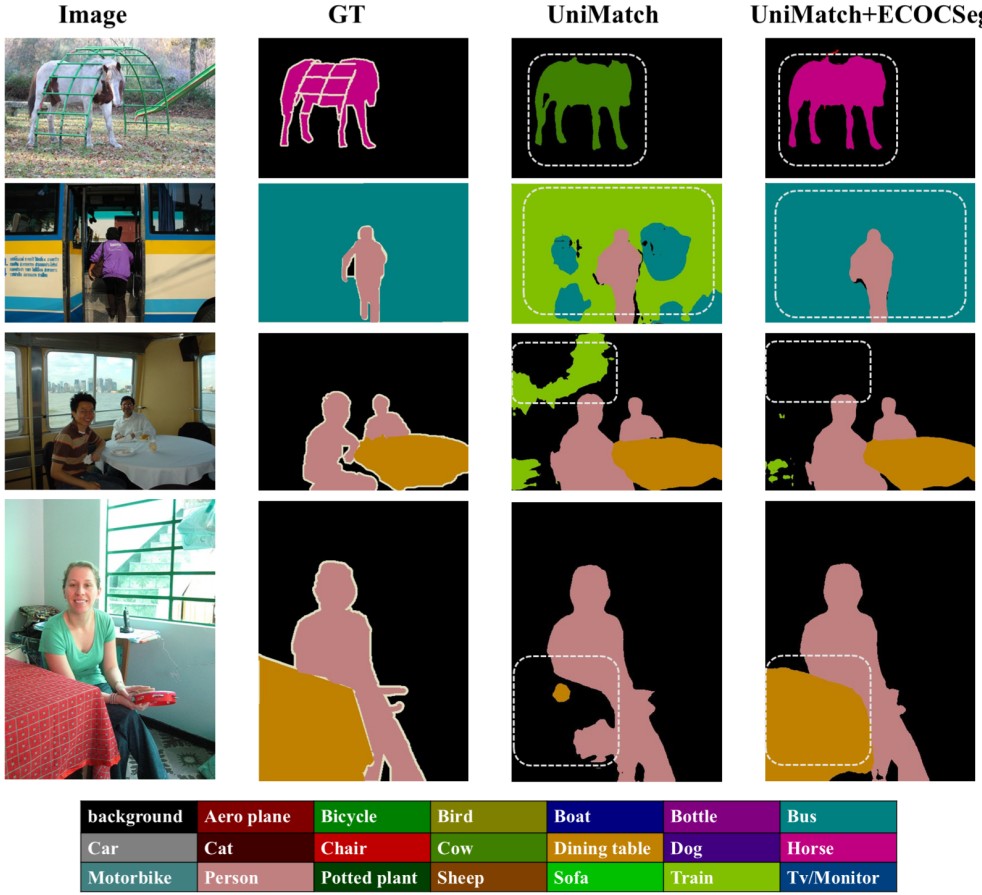

Figure 10: More qualitative comparison built with UniMatch [93] on SSL benchmark of Pascal. The significant improvements are marked with dotted boxes.

