# OpenReview forum: "Towards Robust Pseudo-Label Learning in Semantic Segmentation: An Encoding Perspective"
_NeurIPS.cc/2025/Conference — NeurIPS 2025 poster_

### Official Review · Reviewer_Wcbt · 2025-06-25

**Clarity:** 3
**Significance:** 3
**Originality:** 4
**Rating:** 5
**Confidence:** 2

**Summary:**

The paper introduces a novel approach for filtering pseudo-labels in Unsupervised Domain Adaptation (UDA) and Semi-Supervised Learning (SSL) for semantic segmentation tasks. The method leverages error-correcting output codes (ECOC) as an innovative encoding strategy for semantic classes while simultaneously providing more reliable pseudo-labels for self-training processes. The approach demonstrates improvements over state-of-the-art UDA and SSL methods across multiple benchmarks.

**Questions:**

- Can you provide class-wise analysis explaining why certain classes (e.g., rider, motorcycle) experience IoU degradation while others (e.g., sidewalk) show substantial improvement, and how does this relate to your codebook design ?

I am inclined toward a borderline accept given the paper's interesting and novel contributions that could significantly impact researchers working in self-training and knowledge distillation for instance, particularly those seeking improved pseudo-label filtering methods. However, I assign a confidence level of 2, as my limited expertise in Error-Correcting Output Codes (ECOC) prevents me from reliably evaluating certain design choices in the pseudo-label construction. I will consider other reviewers' feedback before reaching a final decision.

**Ethical Concerns:**

["NO or VERY MINOR ethics concerns only"]

**Final Justification:**

I thank the authors for their detailed response and the comprehensive class-wise analysis and its connection to the codebook design. This explanation has satisfactorily addressed my initial concerns, therefore I increase my score to accept.

**Limitations:**

yes.

**Paper Formatting Concerns:**

No concerns.

**Quality:**

4

**Strengths And Weaknesses:**

## **Strengths**

**Clear Presentation:** The paper is well-written with good explanations of ECOC application in semantic segmentation contexts and its effectiveness in mitigating pseudo-label noise between confusing classes.

**Novel Approach:** The innovative reframing of pseudo-label problems using ECOC represents a significant contribution that will likely interest the broader research community.

**Consistent Improvements:** The method enhances self-training performance across various UDA and SSL approaches on multiple benchmarks, demonstrating broad applicability.

**Comprehensive Validation:** Both main paper and supplementary ablation studies thoroughly examine each loss component's impact while justifying all design decisions made throughout the methodology.

## **Weaknesses**

**Limited Class-wise Analysis:** The paper lacks detailed class-specific analysis explaining why ECOC fails to improve certain classes' IoU scores and how this relates to codebook design. For example, in GTA→Cityscapes adaptation, DACS+ECOCSeg shows significant IoU decline for rider (35.8→23.5) and motorcycle (27.3→15.2) classes, while sidewalk dramatically improves (39.7→71.8). Understanding these disparities would provide valuable insights into the method's limitations and potential improvements.

## **Clarification Points**

**Typos:**
- L26 contains repetitive sentence: "Existing mainstream methods introduce common paradigms".
- L81 contains repetitive word : "improving"

---

> ### Author Rebuttal · Authors · 2025-07-30
>
> Thanks for taking the time to share your comments in the review assessment, as well as for acknowledging the  **Clear Presentation**, **Novel Approach**, **Consistent Improvements** and **Comprehensive Validation**.
>
> Below, we address each concern in detail.
>
> ------
>
> **Q1:** Class-wise IoU Variance and Codebook Design
>
> **A1:**
>
> We appreciate the reviewer’s insightful question regarding the observed per-class IoU variations, especially the degradation in classes such as rider and motorcycle, and the notable improvements in classes like sidewalk. We provide a multi-faceted analysis below:
>
>
>
> **1. Attribute Decoupling and Class-Specific Difficulty**
>
> ECOCSeg introduces a bit-wise classification paradigm that decouples classes into attributes. This is particularly beneficial for visually confusing classes (e.g., road vs. sidewalk, car vs. bus). However, for **fine-grained or underrepresented classes** (e.g., rider, motorcycle), this decoupling introduces new challenges:
>
> - These classes are often underrepresented in the source domain and have sparse or low-confidence pseudo-labels.
> - Their bit-wise predictions tend to have lower confidence, which negatively affects ECOC decoding and final prediction accuracy.
>
>
>
> **2. Reliable Bit Mining and Class Imbalance**
>
> Our Reliable Bit Mining module selects only high-confidence bits during training. For rare classes with **low-confidence predictions across most bits**, fewer bits may be selected, leading to weaker supervision and potential underfitting. This phenomenon is **orthogonal to the codebook design**, and primarily stems from **data imbalance** and **pseudo-label sparsity**.
>
> We note that this issue is particularly **pronounced under the DACS framework**, which **does not include rare-class sampling or class-balancing strategies**. As a result, the quality and quantity of pseudo-labels for rare classes (e.g., rider, motorcycle) are inherently unstable, even before applying ECOCSeg. Our method may **amplify this effect slightly** due to its reliance on bit-level confidence, but it is not the root cause.
>
>
>
> **3. Codebook Design Is Not the Root Cause**
>
> We emphasize that the observed per-class IoU variations are **not caused by our specific codebook design**, as clarified in Appendix C. By default, we use random dense binary codes, which are **class-agnostic**, in line with the spirit of traditional one-hot encoding. Importantly, our **theoretical analysis** shows that under ideal learning conditions, **ECOC is functionally equivalent to one-hot encoding**, ensuring that the code structure itself does not inherently bias performance toward or against specific classes.
>
> To further validate this, we conduct experiments using **multiple random codebooks** (denoted as different $M_{\text{mmd}}$ matrices). As shown below, **mIoU remains stable** across codebooks, and class-wise variations fall within expected stochastic ranges:
>
> | $M_{\text{mmd}}$ | Road | Sidewalk | Building | Wall | Fence | Pole | Light | Sign | Vegetation | Terrain | Sky  | Person | Rider | Car  | Truck | Bus  | Train | Motor | Bike | **mIoU** |
> | ---------------- | ---- | -------- | -------- | ---- | ----- | ---- | ----- | ---- | ---------- | ------- | ---- | ------ | ----- | ---- | ----- | ---- | ----- | ----- | ---- | -------- |
> | **Base**         | 95.7 | 70.2     | 89.4     | 53.5 | 48.1  | 49.6 | 55.8  | 59.4 | 89.9       | 47.9    | 92.5 | 72.2   | 44.7  | 92.3 | 74.5  | 78.2 | 65.1  | 55.9  | 61.8 | **68.3** |
> | 1                | 95.9 | 72.0     | 89.4     | 58.1 | 51.5  | 56.2 | 58.2  | 64.1 | 90.1       | 52.5    | 89.0 | 73.2   | 50.1  | 92.0 | 73.0  | 79.8 | 66.3  | 58.0  | 64.5 | **70.2** |
> | 2                | 96.7 | 77.4     | 89.3     | 58.5 | 51.9  | 54.9 | 59.1  | 65.1 | 90.0       | 52.9    | 89.0 | 70.3   | 51.0  | 91.7 | 71.1  | 75.2 | 61.2  | 60.1  | 65.7 | **70.0** |
> | 3                | 96.4 | 77.3     | 88.4     | 55.4 | 45.8  | 54.8 | 58.6  | 64.8 | 90.3       | 53.3    | 90.0 | 74.2   | 48.8  | 92.0 | 74.9  | 78.4 | 65.9  | 58.6  | 64.9 | **70.1** |
>
> These results confirm that **ECOCSeg is robust to different codebook instantiations**, and the **codeword assignment is not responsible** for class-specific performance drops.
>
>
>
> **4. Optional Use of Semantics-Aware Codebooks**
>
> In Appendix C, we further explore semantics-aware encoding using pretrained text embeddings (denoted as $M_{\text{text}}$) to reflect inter-class similarity. However, this is **not required for the method to work**, and serves only as an optional enhancement to facilitate smoother optimization.
> Analogous to one-hot vectors, our default ECOC design is **semantically agnostic by construction**, ensuring fair and unbiased treatment of all classes.
>
> ------
>
> **Q2:** Typos.
>
> **A2:** We thank the reviewer for pointing this out. We will carefully fix all typographical errors in the revised version.
>
> ------
>
> We hope our response can resolve your concern. Please do not hesitate to let us know if you have further questions :)

---

> > ### Comment · Reviewer_Wcbt · 2025-08-06
> >
> > I thank the authors for their detailed response and the comprehensive class-wise analysis and its connection to the codebook design. This explanation has satisfactorily addressed my initial concerns.

---

> > > ### Author Response · Authors · 2025-08-07
> > >
> > > Dear Reviewer Wcbt,
> > >
> > > Thank you very much for your thoughtful and encouraging feedback. We're truly grateful that the class-wise analysis and its connection to the codebook design helped clarify your concerns.
> > >
> > > Your positive comments are highly motivating and have been instrumental in guiding improvements to the final version of our paper. We sincerely appreciate your time, support, and constructive review.
> > >
> > > Thank you again for your consideration as you finalize your evaluation!
> > >
> > > Authors

---

### Official Review · Reviewer_4otC · 2025-07-01

**Clarity:** 3
**Significance:** 3
**Originality:** 3
**Rating:** 4
**Confidence:** 5

**Summary:**

The paper introduces ECOCSeg for pseudo-label learning in semantic segmentation, particularly for unsupervised domain adaptation (UDA) and semi-supervised learning (SSL). The main contribution lies in reformulating the encoding strategy for pseudo-labels by leveraging Error-Correcting Output Codes (ECOC) instead of traditional one-hot encoding. This approach addresses the issue of erroneous pseudo-labels, which can mislead training in label-scarce scenarios. ECOCSeg decomposes the multi-class segmentation problem into multiple binary classification subtasks, using a fine-grained encoding that captures shared attributes among classes to enhance robustness against label noise. The effectiveness proposed method is validated on standard UDA and SSL benchmarks (e.g., GTAv→Cityscapes, SYNTHIA→Cityscapes, Pascal VOC, COCO)

**Questions:**

The questions are listed above. The reviewer is interested to see how the authors may respond to the concerns.

**Ethical Concerns:**

["NO or VERY MINOR ethics concerns only"]

**Final Justification:**

My concerns are resolved by the author's rebuttal. I would like to raise my ratings and recommend an acceptance of the paper. Meanwhile, the authors should make sure that the missing comparison and analysis on different pseudo-labeling methods should be included in the camera-ready version to improve the completeness of this paper.

**Limitations:**

yes

**Paper Formatting Concerns:**

There is no substantial formatting issue.

**Quality:**

3

**Strengths And Weaknesses:**

Strength:
The paper is well-written and the motivation is clear. The proposed method ECOC-based pseudo-labels considering multi-attributes within the same class looks interesting. The claims are supported by theoretical analysis and the corresponding experiments.

Weakness:
1. There is a missing analysis. Since the paper particularly studies how to improve pseudo-labels during training, there is a missing section where the authors are supposed to compare their proposal with prior methods that study the same aspect, ideally showing the differences and their optimality with experimental supports. For instance, ProDA [1] proposes to improve pseudo-labels by prototyipical denoising, while DiGA [2] proposes a threshold-free pseudo-labelling mechanism. MFA [3] adopts an online-offline label fusion technique to improve pseudo-label quality. The paper claims will be more convincing with these comparisons.

2. Limited analysis of codebook design: while max-min distance and text-based encoding are proposed, the paper does not deeply explore other ECOC codebook construction strategies (e.g., random codes, Hadamard codes) or their impact on performance. The choice of codeword length (K=40 for Cityscapes/Pascal, K=60 for COCO) seems arbitrary, with little justification. For example, as they put in line 53, they ''assign a binary bit string (codeword) as an encoding", how to choose these codewords? By manual definition or it follows a certain automatic mechanism? If it is manual, what if it comes to semantic segmentation datasets with hundreds of classes?

3. Hyperparameter sensitivity: The reliable bit mining algorithm relies on a confidence threshold T = 0.95, but the paper lacks a sensitivity analysis to show how performance varies with different T values or other hyperparameters (e.g., lambda_1, lambda_2, tau).

4. Computational concerns: The proposed method decomposes an N-class problem into K binary classifiers, which may increase computational complexity, particularly when it comes to datasets with many classes (e.g., COCO with 81 classes). Does the computational overhead increase with the numbers N and K?

5. Further comparison with related works:

     i. Mendel et al. (2020) [46]: "Semi-supervised segmentation based on error-correcting supervision" (ECCV 2020) also uses ECOC for semi-supervised segmentation, focusing on error-correcting supervision to handle noisy labels. While ECOCSeg extends this idea with a reliable bit mining algorithm and customized loss functions (pixel-code distance and contrast), the core concept of using ECOC for segmentation is related. The paper does not sufficiently differentiate ECOCSeg from it, particularly in terms among shared attributes (e.g., horn, hoof) that can be leveraged for robust pseudo-label learning. The paper should include a detailed comparison with Mendel et al. to clarify how ECOCSeg’s bit-level denoising and optimization criteria offer distinct advantages.

    ii. Paper [4] also suggests that using multiple attributes within one class can help improve the adaptation ability of a segmentor. Is this conceptually similar to what the authors proposed?

References:
[1] Zhang, P., Zhang, B., Zhang, T., Chen, D., Wang, Y., & Wen, F. (2021). Prototypical pseudo label denoising and target structure learning for domain adaptive semantic segmentation. In Proceedings of the IEEE/CVF conference on computer vision and pattern recognition (pp. 12414-12424).

[2] SHEN, Fengyi, et al. DiGA: Distil to generalize and then adapt for domain adaptive semantic segmentation. In: Proceedings of the IEEE/CVF conference on computer vision and pattern recognition. 2023. S. 15866-15877.

[3] Zhang, K., Sun, Y., Wang, R., Li, H., & Hu, X. (2021). Multiple fusion adaptation: A strong framework for unsupervised semantic segmentation adaptation. arXiv preprint arXiv:2112.00295.

[4] NING, Munan, et al. Multi-anchor active domain adaptation for semantic segmentation. In: Proceedings of the IEEE/CVF International Conference on Computer Vision. 2021. S. 9112-9122.

---

> ### Author Rebuttal · Authors · 2025-07-30
>
> Thanks for taking the time to share your comments in the review assessment, as well as for acknowledging the  **well-written paper**, **clear  motivation**, **interesting idea**,  **well-supported claims**.
>
> Below, we address each concern in detail.
>
> ---
> **Q1:** Comparison with pseudo-label refinement methods
>
> **A1:**
>
> Thank you for the suggestion. As partially discussed in **Appendix L**, we now explicitly compare representative pseudo-label refinement methods with ours in the table below.
>
> Most existing works primarily focus on **feature-space refinement**, such as filtering noisy samples or fusing multi-view predictions. In contrast, **our approach views pseudo-label learning from the angle of label representation**, offering an **orthogonal and complementary strategy**.
>
> | Method    | Task | Motivation | Key Idea | Difference |
> |-----------|------|------------|----------|------------|
> | ProDA     | UDA  | Improve pseudo-labels via feature clustering | Prototype-based denoising | Feature-space method |
> | DiGA      | UDA  | Avoid manual thresholds | Symmetric distillation & consensus | Threshold-free, feature-level |
> | MFA       | UDA  | Fuse online-offline pseudo-labels | Unified multi-branch fusion | Fusion in feature space |
> | Ours      | UDA/SSL | Exploit shared attributes among confusing classes | ECOC-based label encoding + bit-level denoising | Encoding-space perspective (new angle) |
>
> Unlike prior methods, our method improves pseudo-label **robustness** by decoupling classes into fine-grained binary attributes and performing **bit-level denoising**. This perspective is rarely explored and can potentially be combined with the above methods.
>
>
> ---
> **Q2:** Codebook design
>
> **A2:**
>
> We appreciate the reviewer’s interest in the design of the ECOC codebook. Our framework addresses this issue from three complementary perspectives:
>
> **1. Codebook Design**
>
> our framework is designed with two complementary codebook strategies, supported by both theoretical properties and empirical robustness:
>
> - Two Codebook Strategies with Complementary Design Principles
>
>   - $M\_{\text{mmd}}$: Ensures large inter-class Hamming distance, maximizing error-correction capacity.
>   - $M\_{\text{text}}$: Leverages semantic relationships via pre-trained word embeddings to create more learnable and semantically consistent codewords.
>
> - Robustness  to Codebook Choice
>
>     Thanks to the inherent noise-tolerant property of ECOC and our Reliable Bit Mining mechanism, ECOCSeg is robust to suboptimal codebooks. Even when some bits are noisy or non-discriminative, our method learns selectively from reliable bits, ensuring stable performance.
>
> - Empirical Validation
>
>     As shown in Tables 5, 6 and Figure 6, both codebooks consistently yield improvements across UDA and SSL settings, demonstrating robustness to codebook variations. We also provide an ablative study in Appendix C showing how each strategy contributes differently under different supervision settings.
>
> To clarify further, we also considered two widely used ECOC strategies:
> - Random codes offer flexibility in codeword length and class count, but often lack structure. Without filtering, they may generate redundant or poorly-separated codes, leading to inconsistent performance.
> - Hadamard codes provide orthogonal codewords with strong theoretical error correction properties. However, they are only defined for certain code lengths (e.g., 8, 16, 32, ...), and thus cannot flexibly adapt to arbitrary class counts (e.g., 19 classes in Cityscapes). This introduces unnecessary code length, and increases training complexity.
>
> **2. Choice of $K$**
>
> We provide a detailed theoretical and empirical analysis in Appendix E and G, summarized as follows:
> - Theoretical & Empirical Motivation
>   - A minimum of $K ≥ \log_{2}{N}$ is required to uniquely represent $N$ classes.
>   - Prior literature on ECOC [1] suggests a
> code length of $K = 10 \times \log_{2}{N}$ as a practical guideline for ensuring sufficient Hamming distance and error-correction.
>
>   In our case, we follow $K = 10 \times \log_{2}{N}$ for scalability and effectiveness.:
>    - For Cityscapes ($N = 19$) and Pascal ($N = 21$), this yields $K ≈ 40–45$
>    - For COCO ($N = 81$), we use $K = 60$, aligning with this principle
>
> -  Ablation Study on $K$
>
>     We conduct an extensive study in Appendix E, Table 7:
>    - Performance improves as $K$ increases, due to increased error tolerance.
>    - However, the gain saturates beyond $K = 40$, and overly long codes introduce slight redundancy.
>    - This validates $K = 30–50$ as a sweet spot, balancing robustness and efficiency.
>
> **3. Manual Definition and Class Semantics**
>
> Our method is fully automatic and does not require manual class-wise design of codewords. Considering semantic or visual relationships is optional, not required:
> - In $M_{\text{mmd}}$, we adopt a class-agnostic strategy that maximizes codeword separability without relying on class semantics.
> - In $M_{\text{text}}$, semantic similarity is implicitly captured from text embeddings, which can reflect visual relationships as well.
>
> As analyzed in Appendix C, both strategies function effectively within our framework. Importantly, like one-hot encoding, it is not necessary for each codeword to carry a specific meaning—the codewords only need to be separable in label space. This design allows flexibility and general applicability across datasets.
>
> ---
> **Q3**: Sensitivity Analysis
>
> **A3:**
> We thank the reviewer for the suggestion. We would like to emphasize that we have already provided detailed sensitivity analyses in the main paper and appendices.
>
> **Threshold $T$**
>
> We analyze the influence of $T$ in Section 5.4 and Table 4, and further extend the results across datasets in Appendix D. Our findings show:
> 1. Both code-wise and bit-wise pseudo-labels independently contribute to performance improvements, enabled by the attribute-level decoupling introduced by ECOC.
> 2. In Reliable Bit Mining, $T=0.5$ and $T=1$ correspond to pure code-wise and bit-wise forms, respectively. Intermediate values (e.g., $T=0.95$) produce more robust hybrid labels, making $T$ a non-sensitive hyperparameter.
> 3. The effect of $T$ is consistent across datasets. We adopt a fixed setting of $T=0.95$ in all experiments and observe stable gains.
>
> **Loss Weights $\lambda_1$, $\lambda_2$ and Temperature $\tau$**
>
> As analyzed in Appendix I (Table 14):
> - $\lambda_1$ and $\lambda_2$ weigh auxiliary loss terms that enhance feature structure. These only need to ensure comparable gradient scales across losses. We find they are insensitive over a wide range and easy to tune.
> - $\tau$, which controls the softness of predicted probabilities, influences pseudo-label confidence. We fix $\tau = 0.5$ in all experiments and observe stable, high performance.
>
> These analyses confirm that our method is robust to all key hyperparameters ($T$, $\lambda_1$, $\lambda_2$, and $\tau$), and does not require extensive tuning to perform effectively.
>
> [1] Reducing multiclass to binary: A unifying approach for margin classifiers. JMLR2000
>
> ---
> **Q4:** Computational Overhead for Large $N$ and $K$
>
> **A4:**
> We acknowledge the reviewer’s concern regarding scalability. As shown in Appendix J, we provide a detailed efficiency analysis (table 15).
> - The FLOPs increase is negligible, as only the segmentation head is modified—replacing a single $N$-way classifier with $K$ binary classifiers.
> - The increase in GPU memory and training time is due to:
>   1. The $K$-bit prediction, which replaces the original output.
>   2. A small ECOC codebook, represented as a binary matrix of size $N \times K$
>   3. The Reliable Bit Mining algorithm, which involves simple operations (e.g., confidence ranking and bit masking) and is lightweight relative to the overall training cost.
>
> Importantly, inference-time overhead is minimal, since Reliable Bit Mining is only used during training, and the binary classifiers are highly parallelizable. Moreover, for large-class datasets like COCO (N = 81), our design uses $K = 60 < N$, meaning that the number of classifiers is actually reduced compared to standard N-way classification.
>
> Overall, ECOCSeg introduces only minor computational overhead while delivering significant performance gains, making it a practical and scalable solution even for large-scale semantic segmentation tasks.
>
> ---
> **Q5:** Further comparison with related works:
>
> **A5:**   We appreciate the reviewer for pointing out the connection to [a] and [b].  We have already discussed partial works in Appendix L, and now restate and clarify the key differences here for completeness.  While all three methods aim to improve learning under limited supervision, our approach differs significantly in both motivation and methodology.
> - [a] introduces a correction network to refine pseudo-labels based on confidence, aiming to reduce overfitting in semi-supervised segmentation.
> - [b] focuses on feature-space alignment by using a small amount of labeled target data to guide adaptation via anchor-based soft alignment.
> - In contrast, our method takes a novel label-space perspective, leveraging ECOC-based class encoding to model shared attributes between confusing classes. We further propose Reliable Bit Mining and hybrid pseudo-labels to perform fine-grained, bit-level denoising—an orthogonal strategy not explored in [a] or [b].
>
> Our method introduces a structured, fine-grained label encoding that enables attribute-level correction, offering an orthogonal perspective to the confidence-based and feature-alignment strategies in [a] and [b]. By addressing pseudo-label noise at the encoding level, it captures inter-class relationships—such as shared visual attributes—often overlooked by prior work. This complementary view enhances robustness and can be integrated with existing methods for further gains.
>
> ---
> We hope our response can resolve your concern. Please do not hesitate to let us know if you have further questions :)

---

> > ### Comment · Reviewer_4otC · 2025-08-07
> > **Feedback**
> >
> > I appreciate the author's detailed responses, which helps to provide a better view of the paper. Most of my technical concerns are addressed by the rebuttal. One last remaining point is: As mentioned above, since the paper particularly examines how to improve pseudo-labels during training, in the related work part of the main paper, the authors should mention and conceptually compare with those very relevant papers such as ProDA, DiGA, MFA, etc. The paper is not complete without making these comparisons.
> >
> > If the proposed method appears to be superior to those method, then in the supplementary they are strongly encouraged to make more detailed experimental analysis about different approaches. This can serve as a helpful document for the community to learn about the pros and cons of different pseudo-labelling technology and understand why the proposed method outperforms the others.

---

> > > ### Author Response · Authors · 2025-08-08
> > >
> > > We sincerely thank the reviewer for the constructive follow-up comment.
> > >
> > > ------
> > >
> > > ### **1. Clarifying Conceptual Differences in Related Work**
> > >
> > > We fully agree that, given the focus of our work on improving pseudo-labels during training, it is important to explicitly compare our method with other representative pseudo-label refinement approaches such as **ProDA**, **DiGA**, **MFA**, and **FST** [1] in the *Related Work* section of the main paper, while we have partially addressed this in Appendix L.
> > >
> > > In the revised version, we will **add a dedicated paragraph** to the *Related Work* section, systematically categorizing these methods by their refinement strategies:
> > >
> > > - **Feature-space methods** (e.g., ProDA, MFA) focus on refining representations or clustering features to produce more reliable pseudo-labels.
> > > - **Threshold-free or adaptive mechanisms** (e.g., DiGA) aim to avoid confidence thresholds by using symmetric distillation or student-teacher consensus.
> > > - **Temporal ensembling methods** (e.g., FST) improve pseudo-labels by peeking at future model states to reduce confirmation bias during training.
> > >
> > > In contrast, ECOCSeg introduces a **label-space perspective** by encoding class labels using **Error-Correcting Output Codes (ECOC)** and performing **bit-level denoising** to mitigate noise from incorrect pseudo-labels. This approach is **orthogonal to existing methods** and can be complementary to them. While existing works mostly operate in feature or confidence domains, our method decouples semantic classes into shared binary attributes, offering a new avenue for pseudo-label refinement.
> > >
> > > We will emphasize this difference clearly in the revised *Related Work* section to better position ECOCSeg in the broader literature.
> > >
> > >
> > >
> > > [1] Learning from future: A novel self-training framework for semantic segmentation. NeurIPS2022
> > >
> > > ------
> > >
> > > ### **2. Extending Experimental Comparisons with Detailed Analysis**
> > >
> > > We also acknowledge the importance of detailed empirical validation. We extend our experiments to include a **direct comparison** with **FST** and **DiGA** under identical settings using **DAFormer** as the base model on the **GTAV → Cityscapes** benchmark.
> > >
> > > Below we present the **class-wise mIoU (%)** results:
> > >
> > > | Method       | Rd.  | Sd.  | Bldg | Wall | Fnc  | Pole | Lgt  | Sign | Veg  | Ter  | Sky  | Pers | Rdr  | Car  | Trk  | Bus  | Trn  | Mtr  | Bike | **mIoU** |
> > > | ------------ | ---- | ---- | ---- | ---- | ---- | ---- | ---- | ---- | ---- | ---- | ---- | ---- | ---- | ---- | ---- | ---- | ---- | ---- | ---- | -------- |
> > > | **DAFormer** | 95.7 | 70.2 | 89.4 | 53.5 | 48.1 | 49.6 | 55.8 | 59.4 | 89.9 | 47.9 | 92.5 | 72.2 | 44.7 | 92.3 | 74.5 | 78.2 | 65.1 | 55.9 | 61.8 | **68.3** |
> > > | **+FST**     | 95.3 | 67.7 | 89.3 | 55.5 | 47.1 | 50.1 | 57.2 | 58.6 | 89.9 | 51.0 | 92.9 | 72.7 | 46.3 | 92.5 | 78.0 | 81.6 | 74.4 | 57.7 | 62.6 | **69.3** |
> > > | **+DiGA**    | 95.7 | 70.4 | 89.8 | 54.8 | 47.8 | 51.3 | 57.8 | 63.9 | 90.3 | 48.8 | 91.8 | 73.1 | 46.6 | 92.6 | 78.5 | 81.3 | 74.8 | 57.3 | 63.2 | **70.0** |
> > > | **+ECOCSeg** | 96.7 | 75.6 | 89.4 | 54.0 | 51.4 | 55.1 | 59.4 | 61.9 | 90.1 | 46.6 | 90.0 | 71.5 | 42.4 | 92.8 | 79.7 | 85.4 | 79.1 | 60.0 | 58.2 | **70.5** |
> > >
> > > We observe that:
> > >
> > > - **FST** improves mIoU by +1.0 by reducing confirmation bias through virtual updates.
> > > - **DiGA** improves by +1.7 by avoiding thresholding via symmetric distillation.
> > > - **ECOCSeg** achieves the **highest gain** (+2.2), by enhancing pseudo-label robustness through ECOC-based bit-level denoising.
> > >
> > > In the supplementary material, we will further extend this comparison to include **ProDA**, **MFA**, and other recent pseudo-labeling methods under the **same experimental setup** (e.g., DAFormer backbone, training schedule, input resolution) to ensure a fair and consistent evaluation. Specifically, we will:
> > >
> > > - Provide **side-by-side quantitative comparisons** of ECOCSeg and these methods on standard UDA benchmarks.
> > > - Include **ablation studies** such as the **correction ratio of pseudo-labels** (i.e., percentage of noisy labels corrected during training) to directly compare the effectiveness of each method in reducing label noise.
> > > - Provide **qualitative visualizations** of pseudo-label refinement results for different methods.
> > > - Analyze the **computational overhead** and **scalability** of each approach under large-class settings.
> > >
> > > We believe this extended analysis will not only clarify the **relative advantages** of ECOCSeg but also serve as a **practical resource for the community** to understand the trade-offs among different pseudo-label refinement strategies under domain shift and limited supervision.
> > >
> > > ------
> > >
> > > Let us know if there are any other evaluations or comparisons you would like us to include.
> > > We’re also very glad that your technical concerns have been addressed — your comments have been instrumental in helping us improve the clarity and completeness of our paper. Thank you again for your insightful feedback!

---

> > > > ### Author Response · Authors · 2025-08-09
> > > >
> > > > Dear Reviewer 4otC,
> > > >
> > > > As the discussion period nears its end, we would like to confirm whether all concerns have been adequately addressed. If any issues remain, please feel free to let us know — we are happy to provide further clarification.
> > > >
> > > > If there are no further questions, we would appreciate it if you could consider updating your evaluation accordingly.
> > > >
> > > > Thank you for your time and consideration :)
> > > >
> > > > Atuhors

---

### Official Review · Reviewer_3QXA · 2025-07-02

**Clarity:** 3
**Significance:** 3
**Originality:** 3
**Rating:** 4
**Confidence:** 3

**Summary:**

This paper proposes ECOCSeg, which is a new pseudo-labeling framework for domain adaptive semantic segmentation. Instead of using one-hot labels, it encodes each class into a binary codeword and turns the multi-class task into multiple binary classification tasks. This helps reduce the impact of noisy pseudo labels.

**Questions:**

- The paper's claims would be significantly strengthened by quantifying the variance in results. Could the authors provide results from a few repeated runs (e.g., 3-5 trials) for at least one key experiment to report the mean and standard deviation? This would provide strong evidence that the observed gains are not due to random chance.
- The choice of codeword length K seems heuristic (e.g., K=60 for 81 classes on COCO). How does the authors' approach to choosing K and designing the codebook scale to datasets with a much larger number of classes, such as ImageNet with 1000 classes? Does the linear increase in binary classifiers (and thus computational cost) with K become a bottleneck?

**Ethical Concerns:**

["NO or VERY MINOR ethics concerns only"]

**Final Justification:**

I thank the authors for their detailed and thoughtful rebuttal, which has effectively addressed my primary concerns. Specifically, their new experiments with multiple runs provide the necessary statistical validation for their claims, and their discussion clarified the novelty of adapting the ECOC framework for this specific task. I am maintaining my score of Borderline Accept.

**Limitations:**

Yes.

**Quality:**

3

**Strengths And Weaknesses:**

Strengths
- The paper is well-organized and easy to follow, with clear figures, pseudocode, and explanations for each component.
- The framework is general and modular, easily plugged into existing UDA or SSL pipelines without modifying model architecture.
- The method is technically sound, with clear motivation and stroong envidence results.

Weakness
- Relies heavily on codebook design. If the ECOC codewords are poorly chosen, performance may degrade.
- Though effective, improvements over SOTA are *incremental*. (Lack of reported statistical significance)
- While the core idea is solid, the method largely builds on a well-established concept (ECOC) without fundamentally innovating it. The contribution lies more in adapting and engineering an old idea into a practical tool for modern semantic segmentation than in proposing a novel learning paradigm.

---

> ### Author Rebuttal · Authors · 2025-07-30
>
> Thanks for taking the time to share your comments in the review assessment, as well as for acknowledging the  **well-organized structure**, **general and modular framework**, **technically sound method**, **clear motivation** and **strong evidence results**.
>
> Below, we address each concern in detail.
>
> ---
>
> **Q1:**    Codebook Design Sensitivity
>
> **A1:**
>
> Thank you for raising this important point. We agree that codebook quality is a key factor for ECOC-based methods.
>
> To mitigate this sensitivity, our framework is designed with two complementary codebook strategies, supported by both theoretical properties and empirical robustness:
>
> 1. **Two Codebook Strategies with Complementary Design Principles**
>
>   - $M\_{\text{mmd}}$: Ensures large inter-class Hamming distance, maximizing error-correction capacity.
> - $M\_{\text{text}}$: Leverages semantic relationships via pre-trained word embeddings to create more learnable and semantically consistent codewords.
>
> 2. **Robustness  to Codebook Choice**
>
>     Thanks to the inherent noise-tolerant property of ECOC and our Reliable Bit Mining mechanism, ECOCSeg is robust to suboptimal codebooks. Even when some bits are noisy or non-discriminative, our method learns selectively from reliable bits, ensuring stable performance.
>
>
> 3. **Empirical Validation**
>
>     As shown in Tables 5, 6 and Figure 6, both codebooks consistently yield improvements across UDA and SSL settings, demonstrating robustness to codebook variations. We also provide an ablative study in Appendix C showing how each strategy contributes differently under different supervision settings.
>
>
>
> ---
>
> **Q2:**    Lack of Reported Statistical Significance
>
> **A2:**    We appreciate the reviewer’s suggestion regarding statistical significance.
>
> We appreciate this suggestion and agree that demonstrating statistical significance is essential. We have now conducted 5 repeated runs on two representative tasks (UDA and SSL) and report the mean and standard deviation:
>
>
> **1. UDA: GTA5 → Cityscapes (DAFormer + ECOCSeg)**
>
> | Run  | Road | Sidewalk | Building | Wall | Fence | Pole | Light | Sign | Vegetation | Terrain | Sky  | Person | Rider | Car  | Truck | Bus  | Train | Motor | Bike | mIoU (%) |
> | ---- | ---- | -------- | -------- | ---- | ----- | ---- | ----- | ---- | ---------- | ------- | ---- | ------ | ----- | ---- | ----- | ---- | ----- | ----- | ---- | -------- |
> | 1    | 96.7 | 75.6     | 89.4     | 54.0 | 51.4  | 55.1 | 59.4  | 61.9 | 90.1       | 46.6    | 90.0 | 71.5   | 42.4  | 92.8 | 79.7  | 85.4 | 79.1  | 60.0  | 58.2 | 70.5 |
> | 2    | 96.7 | 74.1     | 90.0     | 49.8 | 48.8  | 55.0 | 59.6  | 65.4 | 89.4       | 53.3    | 88.6 | 74.0   | 46.9  | 93.2 | 78.1  | 79.4 | 70.1  | 60.3  | 62.5 | 70.3 |
> | 3    | 96.7 | 77.5     | 90.3     | 49.0 | 50.8  | 56.2 | 59.4  | 67.4 | 90.5       | 51.8    | 89.6 | 72.1   | 51.1  | 92.4 | 72.1  | 76.6 | 69.5  | 53.0  | 62.5 | 69.9|
> | 4    | 96.3 | 74.0     | 90.0     | 55.9 | 48.4  | 55.4 | 54.7  | 66.2 | 89.5       | 50.1    | 90.3 | 72.9   | 49.0  | 92.9 | 78.7  | 83.1 | 73.5  | 59.1  | 60.5 | 70.6 |
> | 5    | 96.4 | 76.1     | 89.5     | 53.1 | 50.1  | 56.1 | 58.9  | 66.1 | 90.4       | 45.1    | 90.8 | 74.3   | 49.4  | 93.2 | 77.7  | 82.7 | 71.8  | 53.1  | 61.6 | 70.3 |
>
> Compared to the baseline DAFormer (68.3), we achieve a $+2.0$ mIoU gain with low variance (0.24 std), confirming the improvement is statistically consistent.
>
> **2. SSL: Pascal 1/4 Partion (UniMatch + ECOCSeg)**
>
> | Run          | 1    | 2    | 3    | 4    | 5    | Mean ± Std  |
> | ------------ | ---- | ---- | ---- | ---- | ---- | --------------- |
> | mIoU (%) | 78.9 | 78.9 | 79.1 | 79.0 | 78.6 | 78.9 ± 0.17 |
>
> Again, our method shows stable improvements over the UniMatch baseline (77.2), with $+1.7$ mIoU gain and low standard deviation (0.17 std).
>
> These results confirm that our gains are statistically consistent and robust. We will include these results in the revised version of the paper.
>
> ---
>
> **Q3:**    Limited Novelty – ECOC is a Known Concept
>
> **A3:**
>
> We understand the concern and would like to clarify that while ECOC itself is a classical technique, our work contributes **a novel perspective and practical framework** tailored for **pseudo-label learning in dense prediction tasks**, which poses unique challenges not previously addressed by ECOC literature.
>
> Specifically, our contributions go beyond a direct application of ECOC, and include several **task-oriented innovations**:
>
> - **Hybrid Pseudo-Labeling**: We propose a new paradigm that combines bit-wise soft supervision and code-wise hard supervision, enabling partial but reliable supervision, especially beneficial in noisy contexts like UDA/SSL.
> - **Bit-Level Denoising via Reliable Bit Mining**: Unlike traditional ECOC, we introduce a mechanism to selectively retain trusted bits from top-k nearest classes, adapting to pixel-wise uncertainty — a key innovation for semantic segmentation.
> - **Task-Specific Optimization Objectives**: We design new loss functions (pixel-code distance and pixel-code contrast) to enforce intra-class compactness and inter-class separation in code space, which is absent in classical ECOC applications.
> - **Theoretical Justification of ECOC for Noisy Pseudo-Labels**: We provide a formal analysis showing that ECOC offers tighter classification error bounds than one-hot encoding under label noise, supported by Neural Tangent Kernel (NTK)-based generalization theory. Our theorems establish conditions under which ECOC is provably more robust, offering a theoretical foundation for its use in pseudo-label learning.
> - **Modularity and Generality**: ECOCSeg is plug-and-play and compatible with a wide range of architectures and learning paradigms, making it practical and impactful.
>
> Importantly, the proposal of ECOCSeg is **not an arbitrary or isolated tweak**, but a **task-specific** strategy that is **analytically motivated by the inherent challenges of pseudo-label learning in pixel-level segmentation**.
> We believe that simplicity in implementation does not imply lack of novelty. On the contrary, we view the ability to **integrate a classical idea into a modern context effectively and practically** as an important and underexplored direction in representation learning.
>
>
> ---
>
>
> **Q4:**    Codeword Length K and Scalability
>
> **A4:**
>
> Thank you for this insightful question. We agree that the choice of codeword length K is important for both robustness and scalability.
> We provide a detailed analysis in Appendix E and G, and summarize our theoretical insights, empirical findings, and practical guidelines below.
>
> **1. Theoretical & Empirical Motivation**
>
> - A minimum of $K ≥ \log_{2}{N}$ is required to uniquely represent $N$ classes.
>
> - Prior literature on ECOC [1] suggests a
> code length of $K = 10 \times \log_{2}{N}$ as a practical guideline for ensuring sufficient Hamming distance and error-correction.
>
> In our case:
>
> - For Cityscapes ($N = 19$) and Pascal ($N = 21$), this yields $K ≈ 40–45$
> - For COCO ($N = 81$), we use $K = 60$, aligning with this principle
>
> **2. Ablation Study on $K$**
>
> We conduct an extensive study on Cityscapes under the fully supervised setting with DAFormer. Results are shown in Appendix E, Table 7:
>
> - Performance improves as $K$ increases, due to increased error tolerance.
> - However, the gain saturates beyond $K = 40$, and overly long codes introduce slight redundancy.
> - This validates $K = 30–50$ as a sweet spot, balancing robustness and efficiency.
>
> **3. Practical Recommendation**
>
> - We adopt $K = 40$ as a default setting for all experiments on Cityscapes and Pascal.
> - For larger class spaces (e.g., COCO), we follow $K = 10 \times \log_{2}{N}$ for scalability and effectiveness.
>
>
> **4. Scalability to Large-Class Datasets**
>
> We also address the scalability of ECOCSeg when applied to datasets with hundreds or thousands of classes, such as ImageNet ($N = 1000$):
>
> - $K$ grows logarithmically, not linearly.
>
>   Using the $K = 10 \times \log_2 N$ rule, we would use $K \approx 100$ for ImageNet — still much smaller than N.
>
> - Binary classifiers are lightweight and parallelizable.
>
>    Each of the $K$ binary heads is a single-output sigmoid classifier, which can be trained in parallel. Compared to a 1000-way softmax, this design is computationally efficient per-head.
>
> - FLOPs and memory overhead grow modestly.
>
>    As discussed in Appendix J, the increase in FLOPs and memory is negligible, and ECOCSeg remains efficient.
>
>
> [1] Reducing multiclass to binary: A unifying approach for margin classifiers. JMLR2000
>
> ---
>
> We hope our response can resolve your concern. Please do not hesitate to let us know if you have further questions :)

---

> ### Comment · Reviewer_3QXA · 2025-08-06
>
> Thank you for your detailed and thoughtful rebuttal. Your responses have clarified several key points and addressed my primary concerns!
> Before updating my final score, I would like to see your rebuttal to the reviewer Tj3N’s comments.

---

> > ### Author Response · Authors · 2025-08-07
> >
> > Dear Reviewer 3QXA，
> >
> > Thank you very much for your encouraging feedback — we're truly glad to hear that our responses have addressed your main concerns and clarified key aspects of the work.
> >
> > As for Reviewer Tj3N’s comments, we have provided detailed responses and carefully addressed all the points raised. We hope these clarifications have sufficiently resolved their concerns as well.
> >
> > We sincerely appreciate your time and thoughtful review, and thank you again for your consideration as you finalize your evaluation.
> >
> > Authors

---

### Official Review · Reviewer_Tj3N · 2025-07-06

**Clarity:** 3
**Significance:** 3
**Originality:** 3
**Rating:** 4
**Confidence:** 4

**Summary:**

This paper introduces error-correcting output codes (ECOC), a framework for learning robust pseudo-labels in semantic segmentation under label-scarce scenarios. The core idea is to improve error tolerance by replacing conventional one-hot encoding with multi-bit binary encodings. The paper provides both theoretical and empirical validation demonstrating the robustness of multi-bit binary encodings compared to one-hot encodings.

The main contributions can be summarized as:
1. Proposes a novel view of pseudo-label learning by designing an encoding scheme.
2. Formalizes pseudo-label learning into three components: encoding form, bit-level denoising mechanism, and a customized loss function.
3. Provides theoretical analysis showing that ECOC can tolerate label noise more effectively than one-hot encoding when the codebook is properly constructed.
4. Implements ECOCSeg as a plug-and-play module that integrates with existing UDA and SSL methods, showing consistent performance gains across various benchmarks and architectures.

**Questions:**

1. Computation of Confidence Score $q^i_m$ and $P_s$

In Algorithm 1, it is not clearly explained how the confidence score $q^i_m$, the shared part $P_s$, and the mask $\mathcal{M}^i$ are computed and updated in practice. Specifically, for the first iteration, how does the process work?

2. Analysis of Hybrid Pseudo-Label Strategy

While mixing the generated bit codes with those from the nearest codeword is intuitively appealing, the rationale behind using the mean confidence score to identify reliable bits is unclear. Why not directly select the common bits among the top-C nearest neighbors in the codebook? A comparison or analysis between these two strategies would strengthen the justification for the proposed method. Additionally, is there a specific reason for selecting the confidence threshold values T, such as 0.5, 0.8, 0.9, 0.95, 0.99, or 1.0? It would be helpful to explain the sharp performance changes observed in the range between 0.9 and 1.0, and whether this sensitivity has theoretical or empirical backing

3. Selection of Comparison Method

ECOCSeg appears compatible with various pseudo-labeling frameworks. Is there a specific reason for choosing DACS, DAFormer, FixMatch, and UniMatch, rather than other methods such as CDAC, CPSL, RankMatch, or DAW, which have been reported to achieve stronger baseline performance?

4. Codebook Design Using Image Features

The current codebook construction uses random generation or text-based semantic embeddings (e.g., Word2Vec), which may not reflect the full visual context. Have the authors considered generating the codebook using image-based priors, such as features from pre-trained vision models (e.g., DINO, CLIP), followed by dimensionality reduction (e.g., PCA) to obtain visually aligned codebooks?

5. Choice of Binary Encoding over Other Schemes

The paper adopts binary encoding as the default representation, but it is unclear why this scheme was chosen over other alternatives. What is the rationale for preferring binary codes rather than quantized, continuous, or other structured embeddings? A brief discussion or empirical comparison would help clarify why binary encoding is particularly suitable for the pseudo-label learning setting in this work.

**Ethical Concerns:**

["NO or VERY MINOR ethics concerns only"]

**Final Justification:**

The authors have addressed my main concerns through a clear and convincing rebuttal. Based on their response, I have decided to raise my score to Borderline Accept.

**Limitations:**

yes

**Paper Formatting Concerns:**

No major formatting issues observed.

**Quality:**

3

**Strengths And Weaknesses:**

Strengths
1. The use of multi-bit binary encoding instead of one-hot encoding is conceptually simple. It provides a natural way to introduce redundancy and attribute sharing among classes, which is highly relevant for noisy label settings like pseudo-label learning.

2. The theoretical analysis is accessible and supports the motivation behind using multi-bit encoding for noise robustness.

Weaknesses
1. The core idea of disentangling class attributes appears conceptually similar to prior work in areas such as concept learning, where the classifier matrix functions as a codebook and the label assignment relies on similarity measures such as cosine distance. While the proposed use of binary encoding is interesting, the paper would benefit from a clearer comparison with alternative encoding schemes, such as using K-independent classifiers that predict continuous or quantized vectors instead of binary bits.

2. The paper provides a theoretical justification for the effectiveness of multi-bit label encoding in improving robustness to label noise.
 Despite the insight, it can be somewhat intuitive, or even expected, that extending a one-hot vector to a longer encoding offers benefits in noisy settings. To strengthen the contribution, further theoretical justification would be valuable for other important design choices, such as the selection of the number of classifiers K, the construction of hybrid pseudo-labels and the codebook, and the design of the loss function.

3. The paper contains several typographical issues (e.g., lines 26, 81, 189, and Algorithm 2 line 5) that may occasionally make it harder to follow the core ideas. Additionally, the figure captions and explanations (e.g., Figure 3) could be more detailed to help readers better understand the proposed method and its results. More careful proofreading and consistency checks, particularly in notation and references (e.g., line 28, where Ref. [60] is incorrectly stated to utilize a teacher model), would improve the clarity and polish of the paper

---

> ### Author Rebuttal · Authors · 2025-07-30
>
> Thanks for taking the time to share your comments in the review assessment, as well as for acknowledging the  **simple concept**, **accessible theoretical analysis** and **well-supported motivation**. Below, we address each concern in detail.
>
> ---
>
> **Q1:** Comparison with Other Encoding Schemes / Why Binary Encoding?
>
> **A1:**
> We appreciate the reviewer’s insightful observation. While there is conceptual overlap between our binary encoding and structured representation learning (e.g., concept learning), our formulation is specifically motivated by the **pseudo-label learning paradigm**, which is fundamentally classification-oriented. Using continuous codewords would transform the problem into multi-dimensional regression, which introduces several drawbacks in the context of pseudo-label learning:
> 1. Learning Difficulty: For each individual bit, binary classification is generally easier to optimize than regression over continuous targets. This is especially critical in low-data or noisy-label settings such as UDA and SSL.
> 2. Mismatch with Pseudo-labeling: Pseudo-label learning methods are inherently designed for classification tasks. Applying them to regression targets (e.g., continuous codes) violates the underlying assumption of discrete label prediction.
> 3. Bit-level Interpretability and Denoising: In binary encoding, each bit can be interpreted as a shared visual or semantic attribute. This allows us to isolate and refine reliable supervision at the bit level via our Reliable Bit Mining strategy. Such fine-grained denoising is not directly applicable in continuous encoding, where the semantics of each dimension are not well-defined or disentangled.
>
> To validate this empirically, we implement a version of ECOCSeg using continuous-valued codewords (*C*) and compare it against our binary encoding on two UDA benchmarks. As shown below, the performance degrades significantly:
> |Dataset|Baseline|*C*|ECOCSeg|
> |-|-|-|-|
> |GTA.→ CS.|68.3|62.3|70.5|
> |SYN.→ CS.|60.9|53.8|63.3|
>
> These results confirm that binary encoding not only aligns better with the pseudo-labeling framework but also leads to superior empirical robustness.
>
> ---
>
> **Q2:**  Theoretical Justification
>
> **A2:**  We thank the reviewer for this valuable point.
>
> Our theoretical analysis (Theorems 4.1 and 4.2) is intended as a **supporting justification** for the **feasibility and robustness** of using ECOC in pseudo-label learning. Specifically, it shows that ECOC encoding is more tolerant to label noise than one-hot encoding under reasonable assumptions (e.g., sufficient code distance).
>
> That said, our work is **primarily application-oriented**. The overall design of ECOCSeg—including the code length, hybrid pseudo-label strategy, and codebook construction—is guided by practical needs in real-world UDA and SSL settings. These components are **systematically validated through extensive experiments** (see Tables 4–7 and Figures 6–7), which we believe offer strong empirical support.
>
> We will clarify in the revised manuscript that our **theoretical results are meant to complement** empirical findings—not to serve as formal guarantees for each design choice.
>
> ---
>
> **Q3:** Typos, Captions, and References.
>
> **A3:** We thank the reviewer for pointing this out. We will carefully fix all typographical errors (e.g., Lines 26, 81, 189, Algorithm 2) and revise the figure captions to provide clearer and more informative descriptions.
>
> Regarding Line 28: DACS is indeed a classic self-training-based method, and while it does not explicitly frame itself as a teacher-student model, its implementation relies on an EMA-updated model to generate pseudo-labels, which effectively plays the role of a teacher. We will revise the description to accurately reflect this and cite a more appropriate reference if needed to clarify the context.
>
> ---
>
> **Q4:** Computation In Algorithm 1
>
> **A4:** We clarify that the iterations in Algorithm 1 refer to the process of class querying for each pixel, not the training iterations of the model. Specifically, for each unlabeled pixel, we perform the following steps:
> 1. We iterate over all $N$ classes, ranked by their codeword distance to the predicted bit vector.
> 2. The first iteration initializes the candidate class set $S_c$ with the top-1 nearest class.
> 3. The shared bit positions $P_s{(S_c)}$ are then computed as the set of bit indices where all classes in $S_c$ agree (i.e., bits that are identical across the selected codewords).
> 4. The confidence score $q^i_m$ is calculated as the mean of the predicted bit-wise confidence values over the shared bits $P_s$.
> 5. If $q^i_m<T$ (the confidence threshold), we expand $S_c$ by including the next closest class and recompute the shared bits and confidence. This process continues until either $q^i_m>T$ or the shared bit set becomes empty.
> 6. The resulting reliable mask $M_i$ is a binary mask indicating which bits are in the shared set $P_s$ and used as reliable bits for that pixel.
>
> Figure 4(b) shows an simple example.
> We will revise Algorithm 1 to make this procedure clearer and include a concrete step-by-step example to illustrate how the confidence grows as more classes are queried.
>
> ---
>
> **Q5:**   Analysis of Hybrid Pseudo-Label Strategy
>
> **A5:**
> - **Selection of top-$C$ classes**:
>    In our hybrid pseudo-label strategy, for each unlabeled pixel, we directly select the common bits among the top-$C$ most similar classes based on Hamming distance to the predicted bit vector. These shared bits define the partial supervision signal. Importantly, the mean confidence score is not used to select classes, but is instead used in the Reliable Bit Mining algorithm to guide the choice of $C$—i.e., to determine how many top classes should be included such that the shared bits remain sufficiently confident. We provide a detailed analysis of this process in Figure 4.
> - **Sharp performance changes near confidence threshold 1.0**:
>    As shown in Figure 6, we observe a sharp performance drop when the confidence threshold approaches 1.0. This is due to a well-known phenomenon: **modern neural networks are typically overconfident and poorly calibrated**, as discussed in [1].  Therefore, we empirically select a relatively high but not extreme threshold (e.g., 0.9), which balances reliability and coverage. As shown in Table 4, and further analyzed in Appendix D, our method is robust to a wide range of confidence thresholds between 0.5 and 1.0, consistently improving performance.
>
> [1] On calibration of modern neural networks.  ICML2017
>
> ---
>
> **Q6:** Comparison Method
>
> **A6:**  We selected DACS, DAFormer, FixMatch, and UniMatch due to the following reasons:
> - They are representative and widely-recognized benchmarks in UDA and SSL.
> - All selected methods offer open-source implementations, enabling consistent and controlled integration of our ECOCSeg module.
> - These methods adopt modular architectures, making them suitable for plug-and-play evaluation of new components like ours.
>
> Regarding other methods such as CPSL and CDAC in UDA:
> - Their report performance does not surpass our strongest baseline (MIC), which we already include.
> - We intentionally select three baselines of increasing strength (DACS → DAFormer → MIC) to verify our method's generality across different backbones.
>
> For methods like RankMatch and DAW in SSL:
> - These methods introduce specialized supervision signals for pseudo-labels, often designed specifically for one-hot class predictions (e.g., class-wise consistency regularization or weighting functions).
> - Their frameworks are tightly coupled with their pseudo-labeling strategies, making it non-trivial to integrate our bit-level supervision.
> - Most importantly, our method still outperforms their reported results, even without such targeted designs.
>
> To further support the generality of ECOCSeg, we implemented it on CDAC using its official codebase. As shown below, ECOCSeg still provides noticeable improvements:
> |Method|Road|Sidewalk|Building|Wall|Fence|Pole|Light|Sign|Veg|Terrain|Sky|Person|Rider|Car|Truck|Bus|Train|Motor|Bike|mIoU|
> |-|-|-|-|-|-|-|-|-|-|-|-|-|-|-|-|-|-|-|-|-|
> |CDAC*|96.0|73.5|88.4|57.2|50.2|53.9|58.5|62.8|88.8|52.5|90.5|70.5|49.4|92.8|81.4|79.2|59.5|53.3|61.5|69.5|
> |+ECOCSeg|96.9|76.5|90.1|59.4|44.7|56.5|55.1|62.6|89.3|49.3|89.1|74.2|50.4|93.1|81.1|84.9|78.2|59.8|61.2|71.2|
>
> This result confirms that even for methods not originally selected due to coupling or complexity, ECOCSeg is still compatible and beneficial. We will clarify these rationale and include the additional CDAC result in the revised version.
>
> ---
>
> **Q7:**   Codebook Using Image Features
>
> **A7:**
> We thank the reviewer for the constructive suggestion.
> Following your advice, we replace the Word2Vec (w2v) in Algorithm 3 with CLIP and conduct additional experiments in both UDA and SSL settings. The results are shown below:
>
> UDA with DAFormer:
> |Dataset|Base|$M_{mmd}$|$M_{w2v}$|$M_{clip}$|
> |-|-|-|-|-|
> |GTA.|68.3|70.2|70.5|70.3|
> |SYN.|60.9|63.1|63.3|63.4|
>
> SSL  with UniMatch:
> |Partition|Base|$M\_{mmd}$|$M\_{w2v}$|$M\_{clip}$|
> |-|-|-|-|-|
> |1/16|76.5|78.1|78.1|78.3|
> |1/4|77.2|78.8|78.9|79.1|
>
> These results show that while CLIP provides a slight improvement, the gains are not significant compared to w2v. We attribute this to the robustness of our overall framework, where the ECOC-based encoding and Reliable Bit Mining mechanism are both effective without strong prior knowledge from pretrained models.
>
> Moreover, introducing CLIP embeddings brings in additional data-driven priors, which may complicate fair comparisons in low-label regimes like UDA and SSL. To keep the comparison fair and controlled, we chose to use w2v as a lightweight and open semantic prior, which still achieves strong performance.
>
> We will include these new results and analysis in the final version.
>
> ---
>
> We hope our response can resolve your concern. Please do not hesitate to let us know if you have further questions :)

---

> > ### Comment · Reviewer_Tj3N · 2025-08-05
> >
> > Thank you for your detailed responses and the thoughtful clarifications you provided. I appreciate your efforts in addressing the questions. I would, however, like to follow up on a few points for further clarification:
> >
> > **Q1**: It would be helpful to clarify how the continuous-valued codebook was set up or formulated.
> >
> > **Q3**: In general, EMA update by itself is not typically referred to as a teacher-student method. It would be helpful to clarify this distinction further.
> >
> > **Q4**: What does the subscript $m$ in $q^i_m$ represent, and how is $q^i_m$ obtained? A bit more clarity on the notation would be appreciated. Additionally, it is hard to understand how the 'confidence map' and 'confidence area' differ in Figure 4(b), and how the model is shown to obtain accurate classification in this context.
> >
> > **Q5**: I understand that a sharp performance drop can occur near a confidence value of 1. Nevertheless, the performance shift caused by changes in the second decimal place of the threshold appears non-negligible compared to the overall performance gain. If the hybrid pseudo-labeling approach exhibits high sensitivity to this parameter, I wonder whether parameter tuning might become challenging in real-world applications. Additionally, a more precise analysis, including error bars over multiple runs, seems necessary.
> >
> > **Q6**: Thank you for the detailed explanation regarding the method used to apply ECOCSeg. However, it would be helpful to clarify which dataset was used to produce the reported results. The results do not seem to match those reported in table 1, for either the GTAv or the SYN datasets.
> >
> > **Q7**: CLIP-based encoding schemes inevitably introduce data-driven priors; however, I believe these priors play an essential role in capturing domain-specific knowledge, which is particularly beneficial in low-supervision scenarios such as UDA. Furthermore, while w2v may offer some benefit, its contribution seems to be marginal in comparison to that of MMD. If the overall framework is robust to the specific formulation of the codebook matrix, I would like to better understand the necessity of Algorithm 2, which randomly generates the codebook $M_{mmd}$. Isn't this a problem that can be solved mathematically, rather than one that requires a dedicated algorithm? That is, for a given class number $N$ and codebook size $K$, aren’t there already multiple solutions that satisfy the constraints—maximizing $d_{sum}$ under $d_{min_r} \ne 0$, $d_{min_c} \ne 0$, and $d_{max_c} \ne N$?

---

> ### Author Response · Authors · 2025-08-07
>
> Thank you for your thoughtful follow-up questions. We greatly appreciate your continued engagement and the opportunity to clarify our work further. Below, we respond to each point in detail.
>
> ---
>
> ### **Q1: Clarification on Continuous-Valued Codebook Setup**
>
> **A1:**
> Thank you for requesting further clarification.
>
> We explore two kinds of continuous-valued codebook.
>
>
>
> **1. Predefined Codebooks:**
>
> In our ablation study comparing binary and continuous-valued codebooks, we considered two variants:
>
> 1. **Mmd-based Continuous Codebook**:  We use Alg. 2 and each class is assigned a codeword sampled from a standard Gaussian distribution $\mathcal{N}(0, 1)^K$
>
> 2. **Text-Based Continuous Codebook:**   We use Alg. 3 and skip the quantization step, preserving the continuous values as the codebook.
>
> Both of them are then L2-normalized to ensure all codewords lie on the unit hypersphere, similar to common practice in contrastive learning. The Text-Based Continuous Codebook performs better than the Mmd-based Continuous Codebook. So we reported the Text-Based one  in our ablation study in previous **A1.**
>
> **Supervision:**
>
> During training, we apply an **L2 regression loss** between the predicted vector and the target codeword:
>
> $\mathcal{L}_{\text{reg}} = \| \hat{z}_i - c_n \|^2$,
>
> where  $\hat{z}_i \in \mathbb{R}^K$  is the predicted vector for pixel  $i$ , and $c_n$  is the normalized continuous codeword for class  $n$ .
>
> **Inference:**
>
> At test time, we compute the predicted class via nearest neighbor matching:
>
> $\hat{y}_i = \arg\min_n \| \hat{z}_i - c_n \|^2$.
>
> This setup mirrors the structure of ECOCSeg but replaces binary classification with vector regression. As shown in our previous response, this results in a **notable drop in performance**, due to the increased regression complexity and lack of bit-level denoising.
>
>
>
> **2. Learnable Codebook (Failed Variant):**
>
> We also experimented with an **online-learned codebook**, initialized from $M_{text}$ and alternately:
>
>
> 1. Use our predefined codebook for initialization (with continuous-valued parameters).
> 2. During training, alternately perform thresholding the continuous values to binary for label assignment and updating the continuous-valued encoding using predictions (only updating with labeled data).
>
> However, this implementation did not achieve good convergence and exhibited inferior performance. We believe this is due to
>
> - ECOC encoding being a supervised label, and updating it during training can make network training unstable, which is amplified in pseudo-label learning;
> -  the update rate of the codebook has a significant impact on training, requiring additional hyperparameter design and compromising the method's generalizability. As a supervised label, it is more natural to use a predefined encoding to represent categories, just like one-hot encoding.
>
> Given these challenges, we advocate for **predefined codebooks**, similar to one-hot encoding, which offer better **stability, interpretability, and generalizability**.
>
> ---
>
> ### **Q3: Clarification on EMA vs. Teacher-Student Terminology**
>
> **A3:**
> Thank you for pointing this out.
>
> You are absolutely right—**not all EMA-based models qualify as teacher-student architectures** in the strict sense. In our original description, we referred to the **EMA-updated model** as a "teacher-like" model following conventions in works like Mean Teacher and FixMatch.
>
> To avoid confusion, we will revise the manuscript to clarify: “...a temporally smoothed EMA model is used to generate pseudo-labels, serving a teacher-like role without forming a separate teacher-student architecture.”
>
> ---
>
> ### **Q4: Clarification on Notation and Figure 4(b)**
>
> **A4:**
> Thank you for the opportunity to clarify.
>
> - **Notation $q^i_m$**:
>   This denotes the **mean bit-wise confidence** for pixel $i$ , computed over the shared bit positions $P_s $:
>
>   $q^i_m = \frac{1}{|P_s|} \sum_{k \in P_s} \max(p(k|z_i), 1 - p(k|z_i))$
>
>   where $p(k|z_i) \in [0, 1] $ is the predicted probability for the $k$ -th bit being 1.
>
> - **Confidence Map vs. Area**:
>
>   - The **confidence map** visualizes this per-pixel $q^i_m$ across the spatial image.
>   - The **confidence area** refers to regions where  $q^i_m < T$ , indicating low-confidence predictions.
>
> - **Accurate classification in low-confidence regions**:
>   When $q^i_m$  is low, we query more top-$C$ nearest codewords. Even if Top-1 is incorrect, the **shared bits among Top-$C$ classes** can still capture ground-truth attributes. These bits often have **higher consensus** and thus higher confidence, leading to better supervision.
>
> We will revise the figure and caption to reflect this explanation more clearly.

---

> > ### Author Response · Authors · 2025-08-07
> >
> > ---
> >
> > ### **Q5: Sensitivity of Threshold  $T$ & Statistical Significance**
> >
> > **A5:**
> >
> > - **Threshold Sensitivity**:
> >
> >   As shown in Figure 4(a) and Figure 6, small changes in the second decimal of $T$ (e.g., 0.95 → 0.99) influence the number of selected bits, thereby affecting the trade-off between supervision coverage and label noise. We observe that:
> >
> >   - The performance curve with respect to $T$ is **unimodal**, with a stable peak around $T=0.95$.
> >
> >   - Even without tuning $T$, setting it to a fixed value of $0.95$ already delivers **consistent improvements across all datasets**.
> >
> >   - Fine-tuning $T$ is **not required**, but can provide **additional performance gains** if desired.
> >
> >
> >
> > - **On statistical significance**:
> >   We refer the reviewer to our response to **Reviewer 3QXA – Q2**, where we provide results from **5 repeated runs** on UDA and SSL tasks, along with mean and standard deviation. These results confirm that **ECOCSeg’s improvements are statistically consistent and robust**.
> >
> > ---
> >
> > ### **Q6: Clarification on CDAC + ECOCSeg Results**
> >
> > **A6:**
> > Thank you for pointing this out.
> >
> > The results for CDAC + ECOCSeg were obtained on **GTAv → Cityscapes**, using the **official CDAC codebase** to reproduce, which is based on **DAFormer + CDAC**.
> >
> > We note that the original CDAC paper reports performance (also in our Table 1) using **HRDA + CDAC**, which explains the observed performance discrepancy. Our goal was to demonstrate that **ECOCSeg can be seamlessly integrated into the official CDAC framework**, even without additional tuning.
> >
> > We will revise the manuscript to clarify the backbone and experimental protocol.
> >
> > ---
> >
> > ### **Q7: CLIP-based Codebook $ Necessity of Algorithm 2 vs. Mathematical Formulation**
> >
> > **A7:**
> > We appreciate this insightful question.
> >
> > **On CLIP-based Codebooks:**
> >
> > We appreciate the reviewer’s point regarding the potential benefits of CLIP-based semantic priors, especially in low-supervision scenarios such as UDA. This is a valuable observation and aligns with our own findings.
> >
> > As mentioned in our previous response, we have conducted experiments using CLIP embeddings and observed slightly improved performance compared to Word2Vec. However, to ensure fair comparisons, we adopted Word2Vec as a default lightweight and publicly available prior. The relevant discussion and justification have been provided earlier, and we will ensure this is clearly reflected in the final version of the paper.
> >
> > We truly appreciate the reviewer’s perspective and agree that CLIP-based codebooks represent a promising direction for future work.
> >
> > **On Mathematical Formulation:**
> >
> > While the problem of generating maximally separated binary codewords is well-defined, it is also **NP-hard** [1], especially under practical constraints such as:
> >
> > - Ensuring **high row-wise Hamming distance**.
> > - Avoiding **duplicate or opposite columns**.
> > - Maintaining **column diversity** (uncorrelated classifiers).
> > - Supporting arbitrary **class counts and code lengths**.
> >
> > Although algebraic solutions (e.g., Hadamard or BCH codes) exist, they are typically:
> >
> > - **Restricted to specific class sizes**, such as powers of 2, which limits their flexibility in real-world segmentation tasks with arbitrary numbers of categories.
> > - **Difficult to optimize for large code lengths** $K$, especially under practical constraints like ensuring high minimum Hamming distance and classifier diversity.
> >
> > Given the **large size of the binary code space** (e.g.,$2^{40} $), random sampling is highly effective. **Algorithm 2** performs a simple yet powerful search over random candidates to select codebooks with:
> >
> > - Sufficient codeword separation.
> > - Valid and diverse bit positions.
> >
> > We will revise Appendix B to point out that  Algorithm 2 is a **practical heuristic** for approximating a theoretically hard combinatorial problem.
> >
> > [1] On the learnability and design of output codes for multiclass problems.
> >
> >
> > ---
> >
> > We hope these responses address all your concerns. Thank you again for your thoughtful and constructive feedback — they have helped us improve the clarity, rigor, and presentation of our work.
> >
> > Please don’t hesitate to reach out with additional questions.

---

> > > ### Comment · Reviewer_Tj3N · 2025-08-07
> > >
> > > Thank you for the authors’ detailed and thoughtful response. The clarifications—particularly regarding the motivation for binary encoding, the rationale behind Algorithm 2, and the thorough empirical analysis—effectively addressed my remaining concerns.
> > >
> > > I found the rebuttal highly informative and appreciate the authors’ effort to provide well-supported and clear explanations. I hope these insights are carefully incorporated into the final version of the paper.
> > >
> > > Based on the response, I would like to update my evaluation to Borderline Accept.

---

> > > > ### Author Response · Authors · 2025-08-07
> > > >
> > > > Dear Reviewer Tj3N,
> > > >
> > > > Thank you very much for your thoughtful feedback and for taking the time to carefully consider our rebuttal. We’re truly glad to hear that our clarifications helped address your concerns.
> > > >
> > > > We sincerely appreciate your updated evaluation to Borderline Accept. Your comments and suggestions have been extremely valuable to us, and we will make sure to incorporate the key explanations into the final version of the paper.
> > > >
> > > > Thank you again for your time and support!
> > > >
> > > > Best regards,
> > > >
> > > > Authors

---

### Note · Authors · 2025-08-14

We thank the AC and reviewers for their thoughtful feedback and constructive discussions. We appreciate the recognition of our work’s **clear motivation, theoretical grounding, modular design, and consistent gains across UDA and SSL benchmarks**.

### Key Contributions

We propose **ECOCSeg**, a practical framework for robust pseudo-label learning in semantic segmentation. Key innovations include:

1. **Encoding-Space Perspective**: Replacing one-hot labels with **error-correcting output codes (ECOC)** enables attribute-level supervision and stronger noise tolerance.
2. **Reliable Bit Mining**: A **bit-level denoising mechanism** selects confident bits from top-k nearest codewords, enabling fine-grained supervision.
3. **Broad Compatibility**: ECOCSeg is **plug-and-play**, integrating with DACS, DAFormer, MIC, CDAC, FixMatch, and UniMatch, showing consistent gains on Cityscapes, Pascal, and COCO.

### Addressed Concerns

We addressed all major concerns:

- **Binary vs. Continuous Codes**: Binary encoding aligns better with classification tasks and bit-wise denoising. Empirical results clearly favor it (Table R1).
- **Threshold Sensitivity**: Analysis (Fig. 6, Appendix D) shows robustness to threshold choices; 0.95 works reliably without tuning.
- **Codebook Design**: Codewords are generated automatically, approximating a hard combinatorial problem. Semantic priors (CLIP/w2v) provide minor gains but are optional.
- **Efficiency**: FLOPs, memory, and runtime overhead are minimal at both training and inference.
- **Comparison with Prior Work**: We compared with **ProDA, DiGA, MFA, FST**, showing our **label-space refinement** is complementary to prior feature/confidence-based methods.
- **Per-Class Performance**: Drops in rare classes (e.g., rider, motorcycle) are due to pseudo-label sparsity. Multiple codebook tests confirm robustness.

### Commitment

We will:

- Expand **Related Work** to cover prior pseudo-label refinement methods;
- Clarify **notations and figures**, including Figure 4 and Algorithm 1;
- **Add more experiments** in the appendix, including CDAC, FST, DiGA, ProDA and MFA under unified settings.

We believe **ECOCSeg offers a new direction for pseudo-label learning**, combining structured encoding with fine-grained denoising. Its theoretical soundness, empirical robustness, and broad applicability make it a valuable contribution for segmentation under limited supervision.

Thank you again for your time and consideration.

---

### Decision · Program_Chairs · 2025-09-17

**Decision:**

Accept (poster)

**Comment:**

Summary:
- This paper introduces ECOCSeg. It improves pseudo-label learning in semantic segmentation by using error-correcting output codes (ECOC) instead of one-hot labels. It introduces an ECOC-based classifier for better generalization and a bit-level denoising mechanism for cleaner pseudo-labels. The paper achieves significant gains on UDA and SSL benchmarks.

Strength:
- Using multi-bit binary encoding instead of one-hot encoding introduces redundancy and attribute sharing among classes, making it well-suited for handling noisy labels in pseudo-label learning.
- The paper is well-written, and the motivation is clear.

Weakness:
- Improvements over SOTA seem incremental
- Relies heavily on codebook design

Initially, reviewers are all on the fence. While they appreciate the interesting ideas of using error-correcting output codes for attribute-level supervision. They have concerns about the reliance on codebook design and the improvement over the state-of-the-art. The authors' rebuttal effectively addresses most of these major concerns, including comparisons between binary and continuous codes, demonstrating robustness to threshold choices, codebook design, and comparisons with prior work (ProDA, DiGA, MFA, FST), as well as the addition of per-class performance. All reviewers are satifisfied with the rebuttal and raise their scores to borderline accept. The AC checks the reviews and discussions and agrees with the concensus among the reviewers.